JCB Journal of Cell Biology

# Kinetochores grip microtubules with directionally asymmetric strength

Joshua D. Larson[1]🄳, Natalie A. Heitkamp[1]🄳, Lucas E. Murray[1]🄳, Andrew R. Popchock[2]🄳, Sue Biggins[2]🄳, and Charles L. Asbury[1]🄳

For accurate mitosis, all chromosomes must achieve "biorientation," with replicated sister chromatids coupled via kinetochores to the plus ends of opposing microtubules. However, kinetochores first bind the sides of microtubules and subsequently find plus ends through a trial-and-error process; accurate biorientation depends on the selective release of erroneous attachments. Proposed mechanisms for error-correction have focused mainly on plus-end attachments. Whether erroneous side attachments are distinguished from correct side attachments is unknown. Here, we show that side-attached kinetochores are very sensitive to microtubule polarity, gripping sixfold more strongly when pulled toward plus versus minus ends. This directionally asymmetric grip is conserved in human and yeast subcomplexes, and it correlates with changes in the axial arrangement of subcomplexes within the kinetochore, suggesting that internal architecture dictates attachment strength. We propose that the kinetochore's directional grip promotes accuracy during early mitosis by stabilizing correct attachments even before both sisters have found plus ends.

## Introduction

The fidelity of mitosis is astounding. The loss rate for a budding yeast chromosome is only $10^{-6}$ per cell per generation (Lacefield et al., 2009). The loss rate for a human chromosome in normal tissue culture cells is typically $<10^{-3}$ per cell per generation (Klaasen and Kops, 2022). Accurate segregation requires chromosomes to become bioriented, with sister chromatids attached via their kinetochores to the plus ends of microtubules emanating from opposite poles of the mitotic spindle. But kinetochores initially attach randomly to microtubules from either pole and they usually bind first to the sides of the filaments (Hayden et al., 1990; Rieder and Alexander, 1990; Tanaka et al., 2005; Itoh et al., 2018), subsequently finding plus ends when external forces transport them to the ends (Tanaka et al., 2005; Kapoor et al., 2006; Torvi et al., 2022), or when side-attached microtubules shorten and bring their disassembling ends to the kinetochores (Inoué and Salmon, 1995; Tanaka et al., 2005). At first, the sister kinetochores often attach erroneously to microtubules from the same pole, in which case the microtubules cannot generate tension to pull them apart. The kinetochores somehow sense this error and, in response, trigger their own detachment to give another chance for proper attachments to form. Conversely, correctly attached sisters come under tension from the opposing microtubules and grip the microtubules stably (Nicklas, 1997; Sarangapani and Asbury, 2014; Lampson and Grishchuk, 2017). This selective stabilization of correct, tension-bearing attachments is the fundamental basis for mitotic accuracy.

Mechanisms proposed to explain the preferential stabilization of tension-bearing kinetochore attachments have focused mainly on plus-end attachments (Sarangapani and Asbury, 2014; Lampson and Grishchuk, 2017; Funabiki, 2019). Aurora B kinase is thought to selectively release plus-end attachments that lack tension, whereas the higher tension on correct plus-end attachments is thought to protect them from Aurora B (Liu et al., 2009; Funabiki, 2019; Lampson and Grishchuk, 2017; Sarangapani and Asbury, 2014). We recently provided direct confirmation that tension by itself suppresses Aurora-triggered detachment of kinetochores from dynamic plus ends (de Regt et al., 2022). But our earlier discovery of an intrinsic catch bond-like behavior, where tension stabilizes kinetochore-tip attachments in the absence of Aurora B kinase (Akiyoshi et al., 2010; Miller et al., 2016, 2019), indicates that additional mechanisms have evolved to promote biorientation. Considering the trial-and-error basis of mitosis, multiple mechanisms are probably required to explain its extremely high fidelity.

Some evidence suggests that side attachments are regulated distinctly from end attachments. Aurora B phosphorylation

[1]Department of Physiology and Biophysics, University of Washington, Seattle, WA, USA; [2]Basic Sciences Division, Fred Hutchinson Cancer Research Center, Seattle, WA, USA.

Correspondence to Charles L. Asbury: casbury@uw.edu; Joshua D. Larson: jdlarson@uw.edu; Sue Biggins: sbiggins@fredhutch.org

C.L. Asbury is the lead contact.



weakens end attachments formed by purified yeast kinetochore subcomplexes in vitro (Flores et al., 2022; Umbreit et al., 2012), but side attachments formed by these same subcomplexes are relatively unaffected (Doodhi et al., 2021). The establishment of end attachment correlates with the release of checkpoint signaling factors from kinetochores, whereas side-attached kinetochores under similar levels of tension retain the signaling factors (Kuhn and Dumont, 2017). Differential regulation of side versus end attachments could help explain the high fidelity of biorientation (Doodhi and Tanaka, 2022), especially if side attachments moving productively toward biorientation were selectively stabilized relative to those moving away from biorientation. However, no evidence for such a directional selectivity has been reported.

The vital importance of plus-end attachments for accurate chromosome segregation suggests that kinetochores might possess an intrinsic, preferential affinity for microtubule plus ends. A classic study has demonstrated that kinetochores on chromosomes isolated from Chinese hamster ovary (CHO) cells preferentially capture the plus ends of microtubules in vitro (Huitorel and Kirschner, 1988), but the molecular basis for this plus-end preference remains unknown, and no such preference has been demonstrated for kinetochores isolated from other cell types.

Here, we show that individual kinetochores assembled de novo in whole budding yeast cell extracts capture microtubules overwhelmingly by their plus ends. Laser trap experiments show that native yeast kinetochores attach to dynamic microtubule tips with substantially higher strength at plus ends than at minus ends. Strikingly, the kinetochores also grip the sides of microtubules with highly direction-dependent strength, indicating an intrinsic sensitivity to the structural polarity of the microtubule wall. A highly direction-sensitive grip is conserved in the human and yeast microtubule-binding Ndc80c subcomplexes. Subdiffraction localization of fluorescent kinetochore proteins indicates that plus end–attached kinetochores are organized with DNA- and microtubule-binding elements separated along the microtubule axis, matching the physiological arrangement during metaphase (Joglekar et al., 2009; Cieslinski et al., 2023). However, side-attached kinetochores adopt a more compact arrangement specifically when they are pulled toward a minus end. These observations suggest that both the plus-end preference and the directionally asymmetric grip of the kinetochore arise from its asymmetric architecture and deformations imposed on its architecture by an external force. We propose that the asymmetric grip of the kinetochore stabilizes its attachment to correctly oriented microtubules specifically during early mitosis, even before both sisters have found plus ends. We also discuss how the kinetochore's asymmetric grip is similar to the directional binding of actin filaments by vinculin (Huang et al., 2017), talin (Owen et al., 2022), and α-catenin (Arbore et al., 2022), behaviors which are thought to promote the self-assembly of organized focal adhesions with appropriately oriented F-actin (Swaminathan et al., 2017; Sun and Alushin, 2023).

## Results

### Individual kinetochores including outer microtubule-binding subcomplexes assemble de novo

We recently showed that the assembly of kinetochores de novo in yeast cell lysates can be directly observed at the single-molecule level using total internal reflection fluorescence (TIRF) microscopy (Popchock et al., 2023). Our approach revealed molecular requirements for the assembly of centromeric nucleosomes carrying the centromere-specific histone H3 variant, Cse4 (CENP-A), which creates the chromosomal foundation for the kinetochore. However, the extent of recruitment of microtubule-binding kinetochore elements and their functional attachment to microtubules remained unexplored.

To measure the recruitment of microtubule-binding elements onto single centromeric DNAs, we tethered 208-bp Atto565-labeled DNAs sparsely onto passivated coverslip surfaces and then incubated them with lysate from a yeast strain harboring an endogenous GFP tag fused to the C-terminus of Ndc80 (Fig. 1 A). After incubation, the lysate was washed away, multicolor TIRF images were collected, and colocalization single-molecule spectroscopy (CoSMoS) was performed (Fig. 1 B) (Friedman and Gelles, 2012; Hoskins et al., 2011; Larson and Hoskins, 2017; Popchock et al., 2023). The fraction of wild type centromeric DNA molecules decorated with Ndc80-GFP was 3.8 ± 2.3% (Fig. 1 C). Analysis of photobleaching suggested that many of these assemblies carried multiple copies of Ndc80-GFP (Fig. S1). We also imaged assemblies after incubation with lysates from strains harboring the inner-kinetochore proteins, Ndc10-GFP (part of the DNA-binding Cbf3 complex), Cse4-GFP (part of the centromeric nucleosome), and Ctf19-GFP (part of the constitutive centromere-associated network) (Fig. 1, A and B). As we previously reported (Popchock et al., 2023), de novo assembly of the inner kinetochore was highly efficient, with 28 ± 1%, 40 ± 1%, and 12 ± 1% of centromeric DNAs recruiting Ndc10-GFP, Cse4-GFP, and Ctf19-GFP, respectively (Fig. 1 C). The more efficient assembly of these inner kinetochore subunits relative to Ndc80-GFP agrees with an ordered hierarchical model of kinetochore assembly and suggests that each step in the assembly process is dependent on the prior recruitment of the innermost kinetochore subunits. In addition to the wild type DNA construct, we tested assembly on a negative control mutant DNA (Fig. S1) containing a 3-bp substitution that eliminates centromere function in vivo (Lechner and Carbon, 1991; Sorger et al., 1994, 1995) and in vitro (Lang et al., 2018). No more than 0.14% of these negative control mutant DNAs colocalized with GFP-tagged kinetochore components (Fig. S1). These observations confirm the specificity of our single-molecule kinetochore assembly assay and demonstrate that about 1 in 25 centromeric DNAs recruits microtubule-binding elements. Considering that many hundreds of individual DNAs can be observed in a single field of view, this level of efficiency was sufficient to enable functional, microtubule-binding behaviors of the individual kinetochore assemblies to be studied.

### Assembled kinetochores capture microtubules with a strong preference for plus ends

To test for microtubule-binding activity, we introduced taxol-stabilized Alexa Fluor 647–labeled microtubules after assembling

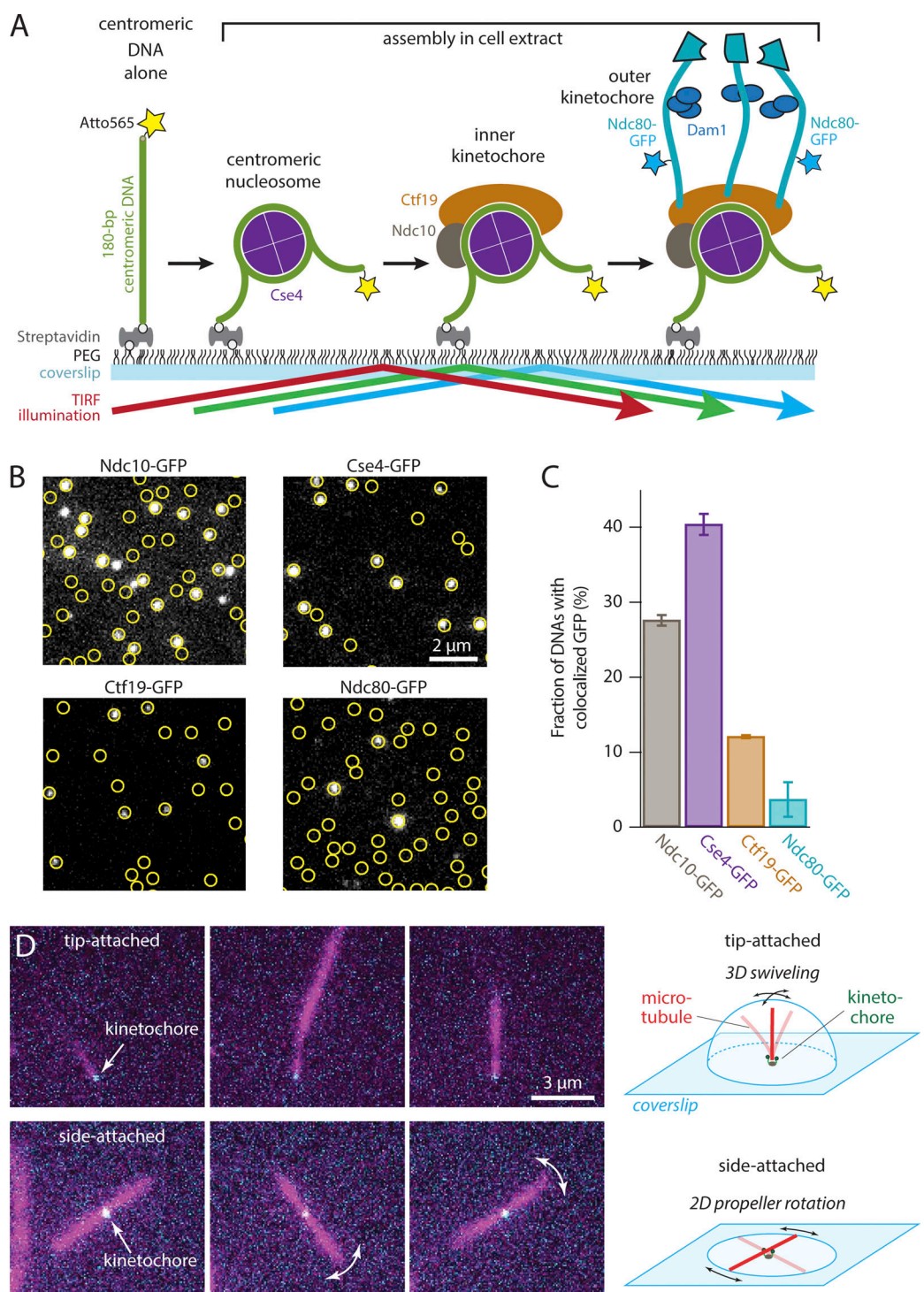

Figure 1. **Individual kinetochores assembled de novo onto centromeric DNAs capture microtubules. (A)** Schematic of the in vitro kinetochore assembly assay. Individual Atto565-labeled centromeric DNAs were tethered sparsely onto a PEG passivated coverslip surface through biotin-avidin linkages. The surface-tethered DNAs were then incubated for 60 min with yeast whole-cell lysate derived from strains with GFP-tagged kinetochore components (Ndc10, Cse4, Ctf19, or Ndc80). Kinetochores assembled spontaneously onto the centromeric DNAs and were then imaged with TIRF microscopy after washing out the extract. **(B)** Kinetochore subcomplexes colocalized with wild type centromeric DNAs. Locations of Atto565-labeled centromeric DNAs (yellow circles) were mapped onto images of GFP-tagged kinetochore subcomplexes (white spots). Scale bar, 2 μm. **(C)** Percentages of centromeric DNAs that colocalized with a GFP signal from indicated kinetochore proteins. Bars show average colocalization ± SEM calculated from $N > 3,400$ DNAs for each kinetochore component from at least nine fields of view recorded across three independent experiments. **(D)** Assembled Ndc80-GFP kinetochores (cyan) readily captured Alexa647 microtubules (magenta) by their tips (top row of images), and sometimes by their sides (bottom row). Tip-captured and side-captured microtubules were easily distinguished by the relative locations of the kinetochore GFP spots and by the Brownian movement of the filaments. The distal ends of tip-captured microtubules swiveled freely in three dimensions (3D), exploring a hemispherical space above the coverslip. Side-captured microtubules mainly rotated in a two-dimensional (2D) plane parallel to the coverslip in a propeller-like fashion.

kinetochores and washing away the lysate. The microtubules were incubated with the kinetochore assemblies for 15 min, excess unbound microtubules were washed away, and the assemblies were then imaged (Fig. 1 D). The individual assembled kinetochores readily captured single microtubules. Capture was specific to the kinetochore assemblies and did not occur in negative controls with non-functional mutant centromeric DNAs. The kinetochore assemblies often captured microtubules by their tips (Fig. 1 D and Video 1), which nearly always colocalized with the fluorescence from Ndc80-GFP. In this tip-attached arrangement, the distal ends of the microtubules swiveled freely by Brownian motion, exploring a hemispherical space above the coverslip. Some kinetochore assemblies captured microtubules by their sides. The Brownian movement of these side-captured microtubules was more restricted. Rotation occurred mainly in a plane parallel to the coverslip, swiveling in a propeller-like fashion (Fig. 1 D and Video 2) with Ndc80-GFP located at the axis of rotation. While the vast majority of captured microtubules had a colocalized Ndc80-GFP signal, in rare instances a captured microtubule appeared to lack Ndc80-GFP. These rare observations could potentially be due to photobleaching or GFPs that had not matured. Alternatively, they might represent capture via the chromosomal passenger proteins, Bir1 and Sli15, which were previously shown to form centromere–microtubule attachments in vitro that do not depend on Ndc80 (Sandall et al., 2006). In any case, our observations show that individual de novo assembled kinetochores can capture the tips and the sides of microtubules, and they demonstrate flexibility in the tethering of the kinetochores to the coverslip.

Given how vital plus-end attachments are for mitosis and that kinetochores on isolated CHO cell chromosomes preferentially capture plus ends (Huitorel and Kirschner, 1988), we hypothesized that the yeast kinetochore assemblies might likewise possess an intrinsic, preferential affinity specifically for microtubule plus ends. To test for preferential plus-end binding, we generated polarity-marked GMPcPP-stabilized microtubules by growing dimly fluorescent plus-end extensions from brightly fluorescent seeds (Howard and Hyman, 1993; Roostalu et al., 2011). We then assembled surface-tethered kinetochores, incubated them with the polarity-marked microtubules, and quantified the fraction of microtubules that were captured by plus versus minus ends (Fig. 2 A). For clear viewing, we applied a gentle flow of buffer (0.6 ml·min⁻¹) to keep the kinetochore-attached microtubules parallel to the coverslip and in the plane of focus. More than 82% of tip-bound microtubules (162 out of 196 microtubules examined across eight technical replicates) were captured by their plus ends (Fig. 2 A and Fig. S2). This preference was observed in the absence of motor proteins (Lang et al., 2018), ATP, and microtubule dynamics, implying that kinetochores themselves have a strong intrinsic affinity for features specific to microtubule plus ends.

**Plus-end attachments support more tension than minus-end attachments**

Kinetochores sustain tension almost continuously once they are properly end-attached in vivo (Waters et al., 1996), so their load-bearing capacity is important for function. We therefore wondered whether the plus-end binding preference uncovered in our TIRF experiments would affect a kinetochore's load-bearing capacity. Using a laser trap, we measured the rupture strengths of native kinetochore particles isolated from yeast lysate via immunoprecipitation (Akiyoshi et al., 2010). As in our prior work, the native kinetochore particles were conjugated sparsely to polystyrene microbeads and then attached near the tips of individual dynamic microtubules growing from coverslip-anchored seeds. After an initial preload tension of ∼1 pN was applied to slide a kinetochore-bead to the end of a microtubule, the pulling force was gradually increased (at 0.25 pN·s⁻¹) until the kinetochore bead detached from the microtubule (Fig. 2 B). Plus and minus ends were readily identifiable in these experiments because the plus ends grew faster, extending farther from the coverslip-anchored seeds than the minus ends. The distribution of rupture strengths measured at plus ends was very similar to our previous measurements (Akiyoshi et al., 2010; Miller et al., 2016), with a mean strength of 9.7 ± 1.0 pN (mean ± SEM from $N$ = 43 plus-end attachments) (Fig. 2 C, left). Strengths measured at minus ends were substantially weaker, with a threefold lower mean strength of only 3.3 ± 0.5 pN ($N$ = 26 minus-end attachments; $P = 2 \times 10^{-5}$, based on a Kolmogorov-Smirnov test). In many instances, it was possible to sequentially measure the rupture strength of a single kinetochore-decorated bead at both ends of a microtubule. Irrespective of the order of these measurements, minus end first or plus end first, the strength was always higher at the plus end (Fig. 2 C, right).

**Kinetochores grip microtubule sides with direction-dependent strength**

Before achieving proper plus-end attachments in vivo, kinetochores initially bind to the sides of microtubules (Hayden et al., 1990; Rieder and Alexander, 1990; Tanaka et al., 2005; Itoh et al., 2018). This physiological behavior was also seen in our laser trap experiments with isolated kinetochores. Modest amounts of laser trap tension, 0.5–3 pN, caused side-attached kinetochores to slide toward the ends. We noticed during our rupture strength measurements that side-attached kinetochores often detached from microtubules when sliding toward minus ends, whereas detachment seemed less likely when sliding toward plus ends. This observation, together with the large strength difference between plus versus minus-end attachments, led us to hypothesize that side attachments might be sensitive to the structural polarity of the microtubule wall. To test this idea quantitatively, we used laser trapping to measure the friction between side-attached kinetochores and microtubules (Bormuth et al., 2009; Forth et al., 2014).

Microbeads decorated sparsely with native kinetochore particles were attached to the sides of individual, dynamic microtubules, growing (as described above) from coverslip-anchored seeds. Constant pulling forces between 0.5 and 3 pN were applied parallel to the microtubule axis, and the speeds at which the kinetochore-decorated beads slid along the microtubule were quantified. The direction of force was periodically reversed to assess friction in both directions relative to microtubule polarity (Fig. 2, D and E; and Fig. S3 A). For every

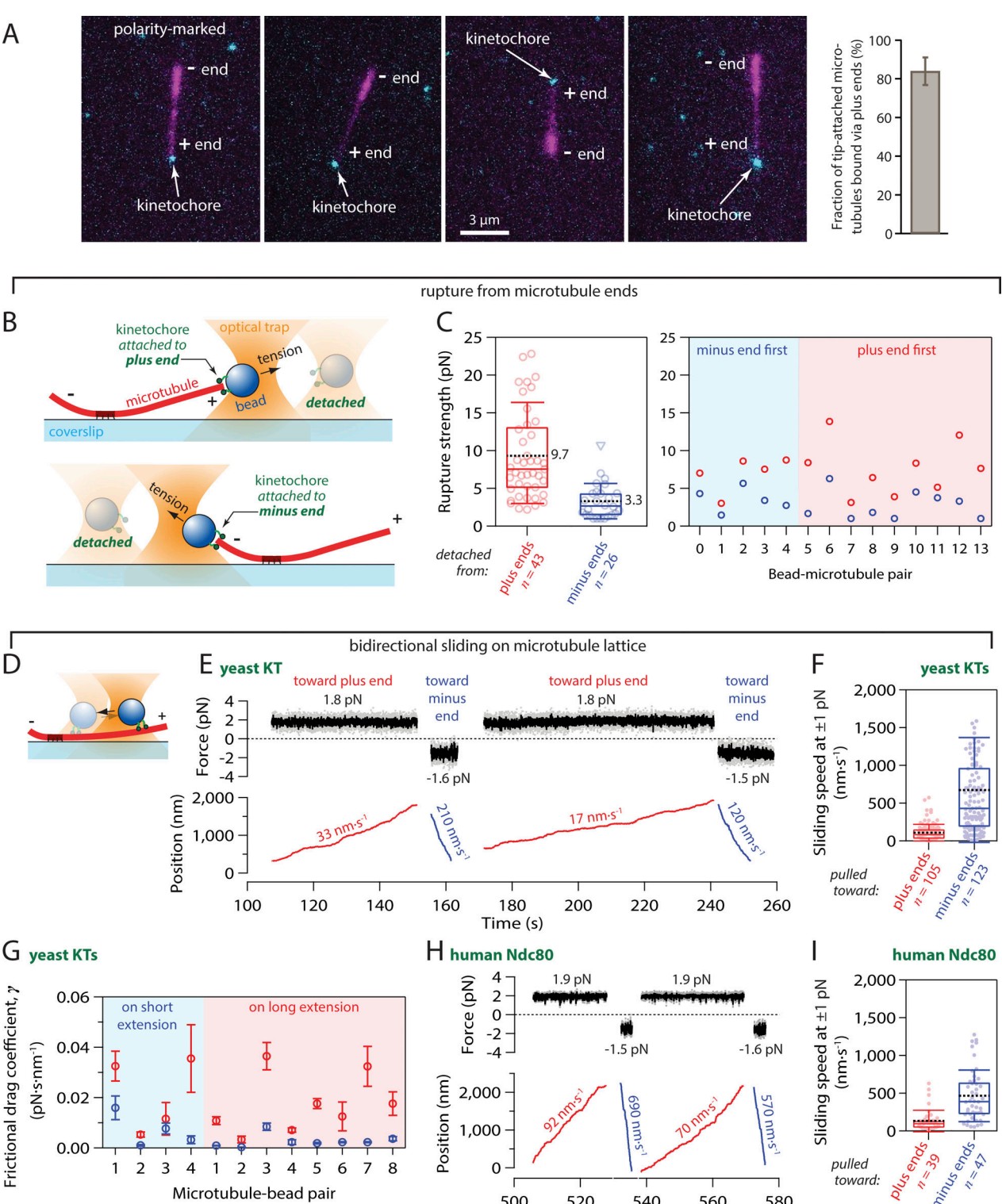

Figure 2. **Kinetochores specifically capture plus ends and grip more strongly when pulled toward plus ends. (A)** Tip-captured, polarity-marked microtubules (magenta), polymerized with dim plus ends and bright minus ends, nearly always bound assembled Ndc80-GFP kinetochores (cyan) by their plus ends. The bar graph shows the percentage of tip-attached microtubules that were bound by their plus ends (mean ± SD from $N$ = 8 experiments examining a total of 196 tip-captured microtubules). **(B)** Schematic of rupture force assay. Native kinetochore particles isolated from yeast were conjugated sparsely to polystyrene microbeads. A laser trap was used to attach a kinetochore bead to either the plus or minus end of an individual dynamic microtubule and then to measure the rupture strength of the attachment. **(C)** Left: Distribution of rupture strengths for yeast kinetochores attached to plus and minus ends ($N$ = 43 and 26 events, respectively). Open circles represent individual strength measurements. Triangles show censored data when rupture strength exceeded the maximum force of the trap or the microtubule broke away from the coverslip surface. Boxes extend from first to third quartiles with medians indicated by central horizontal solid lines. Whiskers extend ± one SD from means, which are indicated by dashed black lines. Right: Rupture strengths for individual

kinetochore beads measured sequentially at both ends of the same microtubule, either minus end first or plus end first as indicated. **(D)** Schematic of bidirectional sliding assay. **(E)** Record of force and position versus time for a kinetochore-decorated bead (yeast KT) attached to the side of a coverslip-anchored microtubule and pulled alternately toward the plus (red trace) and minus end (blue trace). Additional records are shown in Fig. S3. **(F)** Distribution of bidirectional sliding speeds for side-attached yeast kinetochores (KTs) measured at 1 ± 0.5 pN of force applied toward plus and minus ends (N = 105 and 123 events, respectively). Dots represent the speeds of individual sliding events. Boxes extend from first to third quartiles with medians indicated by central horizontal solid lines. Whiskers extend ± one SD from means, which are indicated by dashed black lines. **(G)** Kinetochore beads were tested on both long and short extensions, to confirm that the speed differential arises from microtubule polarity rather than asymmetric anchorage of the microtubule to the coverslip. Frictional drag coefficients for individual kinetochore beads sliding toward plus (red symbols) and minus ends (blue symbols) on short and long microtubule extensions, as indicated. Beads 1–4 were measured sequentially on both extensions of the same microtubule. Symbols represent mean frictional drag coefficient ± SEM (from N > 5 sliding events per bead–microtubule pair). **(H)** Record of force and position versus time for a bead coated with human Ndc80 complex attached to the side of a coverslip-anchored microtubule and pulled alternately toward the plus (red trace) and minus end (blue trace). **(I)** Distribution of bidirectional sliding speeds for human Ndc80c-coated beads measured at 1 ± 0.5 pN of force applied toward plus and minus ends (N = 39 and 47 events, respectively), plotted as in panel F.

bead–microtubule pair examined, sliding toward the plus end was markedly slower than toward the minus end. Under 1 pN of laser trap tension, the average sliding speeds toward plus versus minus ends differed sixfold (109 ± 11 nm·s$^{-1}$ versus 674 ± 62 nm·s$^{-1}$; mean ± SEM from N = 105 and 123 sliding events, respectively, across 24 microtubule-bead pairs; P = 7·10$^{-20}$ by Kolmogorov-Smirnov test) (Fig. 2 F). To compare friction across many bead–microtubule pairs, measured at different forces and on plus- and minus-end extensions, we divided the applied force, F, by the mean sliding speed, $v$, to compute a frictional drag coefficient (Bormuth et al., 2009), $\gamma = F v^{-1}$. The frictional drag during plus-end-directed sliding was uniformly higher, independent of whether beads were tested on the shorter microtubule extensions (where plus-end-directed sliding was toward the coverslip-anchored seeds) or on the longer extensions (where plus-end-directed sliding was toward free microtubule ends) (Fig. 2 G). This control confirms that the speed difference arises from microtubule polarity, rather than from asymmetric anchorage of the microtubule to the coverslip. Altogether, these observations indicate that kinetochores grip the sides of microtubules with a strength that differs markedly depending on the direction of applied force relative to the polarity of the microtubule substrate.

### Yeast and human Ndc80 complexes grip microtubule sides with direction-dependent strength

The primary microtubule-binding kinetochore element, Ndc80c, binds microtubules partly through a stereospecific "footprint," which presumably cannot twist or rotate without breaking its bond to the filament. The stalk of Ndc80c emerges from the "foot" (from the calponin-homology domains) with a tilt toward the microtubule plus end (Cheeseman et al., 2006; Alushin et al., 2010; Muir et al., 2023). Based on this local structural asymmetry, we hypothesized that Ndc80c alone might exhibit asymmetric mechanical behavior similar to the native kinetochore particles.

To determine if the Ndc80 complex by itself grips microtubules with direction-dependent strength, we purified recombinant yeast and human Ndc80c (Flores et al., 2022; Powers et al., 2009; Umbreit et al., 2012; Hamilton et al., 2020; Helgeson et al., 2018) and then measured the speeds at which microbeads coated with each complex slid along the sides of microtubules under plus- and minus-end-directed forces. Because these experiments with recombinant Ndc80c were not

conducted under single-molecule conditions, many complexes on each bead were presumably contacting each individual microtubule tip (Hamilton et al., 2020). This arrangement mimics the physiological situation, where multiple Ndc80 complexes form a multivalent attachment to each kinetochore-attached microtubule. Microtubule attachments based on yeast Ndc80c alone are relatively weak compared with those based on native yeast kinetochores (Akiyoshi et al., 2010; Powers et al., 2009). Consequently, it was challenging to record micrometer-long, bidirectional events during which a bead coated with yeast Ndc80c remained persistently and unambiguously associated with the microtubule. Nevertheless, when such events were recorded, asymmetry was clearly evident (Fig. S3 B). Compared with the yeast complex, human Ndc80c forms stronger, more persistent attachments, enabling the recording of many bidirectional sliding events (Fig. 2 H and Fig. S3 C) and revealing very significant asymmetry (Fig. 2 I). Under 1 pN of tension, the average sliding speeds for human Ndc80c-coated beads pulled toward plus versus minus ends differed almost fourfold (130 ± 23 nm·s$^{-1}$ versus 464 ± 50 nm·s$^{-1}$; mean ± SEM from N = 39 and 47 sliding events, respectively, across 24 microtubule–bead pairs; P = 6·10$^{-9}$ by Kolmogorov-Smirnov test). These observations demonstrate that direction-sensitive grip strength is an intrinsic behavior conserved in both the yeast and human Ndc80c subcomplexes.

### Assembled kinetochores recapitulate in vivo architecture when attached to microtubule plus ends

To further investigate the basis for direction sensitivity, we sought to examine the architecture of individual kinetochore assemblies. When kinetochores are properly attached to microtubule plus ends in vivo, their molecular components are spatially organized (Joglekar et al., 2009; Wan et al., 2009; Dumont et al., 2012). Fibrillar Ndc80 complexes align with the microtubule axis and their "outer" microtubule-binding domains project distally toward the minus ends and spindle poles. Conversely, DNA-binding "inner" kinetochore elements are proximal to the chromosome, oriented toward the microtubule plus ends. To examine the configuration of kinetochores assembled de novo, we mapped the relative positions of various fluorescent-tagged kinetochore components along the microtubule axis by locating their centers of fluorescence. GFP-tagged kinetochores were assembled onto Atto565-labeled DNAs and exposed to taxol-stabilized, Alexa Fluor 647–labeled microtubules. A gentle flow

of buffer (0.6 ml·min⁻¹) was applied with a syringe pump to exert sub-piconewton viscous forces that aligned kinetochore-attached microtubules with the plane of the coverslip (Fig. 3 A). We oscillated the flow direction, causing the microtubules to flip back and forth, reorienting their long axes by 180° with each reversal of the flow (Fig. 3. B and C; and Video 3). The attached kinetochore assemblies were also reoriented together with the microtubules, allowing us to measure distances from the fluorescent-tagged kinetochore components to the tether point on the coverslip with nanometer accuracy. Initially, we focused on kinetochore assemblies that had captured microtubules by their ends.

When a kinetochore assembly periodically reoriented with the flow, the fluorescent marker on its centromeric DNA was displaced from the tether point on the coverslip by 16 ± 1 nm on average (Fig. 3, D and E; mean ± SEM from $N$ = 74 measurement intervals across 13 end-attached kinetochore assemblies). This distance is much shorter than would be expected for a straight B-form DNA helix of ∼200 bp (∼60 nm), presumably because the centromeric DNA, after kinetochore assembly, was tightly wrapped around a centromeric nucleosome. Consistent with this interpretation, the histone H3 variant Cse4-GFP was located 12 ± 1 nm from the tether (mean ± SEM, $N$ = 67 intervals, 9 kinetochores), very close to the centromeric DNA marker. The microtubule-binding component Ndc80-GFP was 37 ± 1 nm from the tether ($N$ = 116 intervals, 19 kinetochores), implying that its GFP tag was located ∼25 nm outward from the nucleosome (Fig. 3, D and E). Considering where the C-terminal GFP tag falls within the structure of the Ndc80 complex (Ciferri et al., 2008; Wei et al., 2005; Zahm et al., 2023), this 25-nm distance suggests that the Ndc80c fibrils were well aligned with the microtubule axis, as they are in vivo (Joglekar et al., 2009). Dam1-GFP, a component of the outer, microtubule-binding Dam1 complex, was located 65 ± 2 nm from the tether ($N$ = 128 intervals, 22 kinetochores) (Fig. 3 E and Fig. S4). This relatively large distance further suggests the axial alignment of Ndc80c fibrils because the Dam1 complex binds nearer to the N-terminus of Ndc80 (Flores et al., 2022; Kim et al., 2017). The implied intrakinetochore separation between the C-termini of Ndc80 and Dam1 was 28 nm, a distance indistinguishable from the intrakinetochore separation measured previously in budding yeast cells during metaphase (Joglekar et al., 2009). We note that force was imposed in our in vitro mapping experiments only by the flow acting on taxol-stabilized microtubules, whereas the polymerization and depolymerization dynamics of attached plus ends in vivo might impose different configurations onto the kinetochore. Nevertheless, our observations show that when de novo–assembled kinetochores are attached to taxol-stabilized plus ends, they are spatially organized in a configuration that closely resembles the molecular arrangement during metaphase in vivo, with DNA-binding subcomplexes proximal to the chromatin and microtubule-binding subcomplexes projecting distally toward minus ends (Joglekar et al., 2009; Cieslinski et al., 2023; Virant et al., 2023).

### Kinetochore architecture is less organized when attached to the sides of microtubules

The molecular organization of side-attached kinetochores has scarcely been explored. By mapping the relative positions of centromeric DNA and Ndc80-GFP within kinetochore assemblies that captured microtubules by their sides, we were able to examine the molecular arrangement of side-attached assemblies and compare them directly to end-attached assemblies, often measured simultaneously on the same coverslips. We focused on assemblies that captured the sides of taxol-stabilized microtubules in an off-center arrangement (Fig. 4, A and B), where the two microtubule segments extending away from the kinetochore had unequal lengths. The longer segment experienced higher viscous drag forces and therefore oriented reliably downstream in the flow (Video 4). Axial positions of fluorescent-tagged centromeric DNA and Ndc80-GFP within these side-attached kinetochore assemblies were tracked in the same manner as for end-attached assemblies (Fig. 4 C). The distribution of distances between the centromeric DNA marker and the tether point on the coverslip was indistinguishable from that measured for end-attached kinetochore assemblies (Fig. 4, D and E; P = 0.15, based on a Kolmogorov-Smirnov test with $N$ = 32 and 74 intervals from 13 side- and 13 end-attached assemblies, respectively). The distribution of Ndc80-GFP distances, however, was wider in comparison to the end-attached assemblies and apparently bimodal, including an elongated subpopulation with a mean distance of 39 ± 1 nm and a compact subpopulation, much closer to the tether, with a mean distance of only 18 ± 2 nm (± SEM, $N$ = 54 and 33 intervals, respectively, 13 side-attached kinetochores) (Fig. 4 E). This observation shows that the molecular architecture of side-attached kinetochore assemblies differs from end-attached kinetochore assemblies, with Ndc80 fibrils often less well aligned to the microtubule axis.

### Side-attached kinetochores are more compact specifically when pulled toward minus ends

Depending on which end of the microtubule was oriented downstream in the flow, the side-attached kinetochore assemblies experienced sub-piconewton viscous pulling forces directed either toward the minus end or toward the plus end. We hypothesized that the two subpopulations, with elongated or compact Ndc80-GFP, might correspond to these two different pulling directions. To test this idea we repeated the Ndc80-GFP distance measurements using polarity-marked GMPcPP-stabilized microtubules. When side-attached kinetochore assemblies were pulled toward minus ends, Ndc80-GFP was closer to the tether, at a distance of only 14 ± 2 nm (mean ± SEM, $N$ = 30 intervals, 11 kinetochores), and when they were pulled toward plus ends, Ndc80-GFP was farther from the tether, at a distance of 39 ± 4 nm ($N$ = 45 intervals, 7 kinetochore assemblies; P = 10⁻⁶ based on a Kolmogorov-Smirnov test) (Fig. 4 F). These measurements demonstrate that the direction of external force directly influences kinetochore architecture, with Ndc80 fibrils adopting a more compact arrangement specifically when the kinetochore is pulled toward the minus end and with Ndc80 fibrils more aligned to the microtubule axis when the kinetochore is pulled toward the plus end.

## Discussion

The directionally asymmetric grip of the kinetochore suggests a previously unrecognized mechanism for promoting accuracy

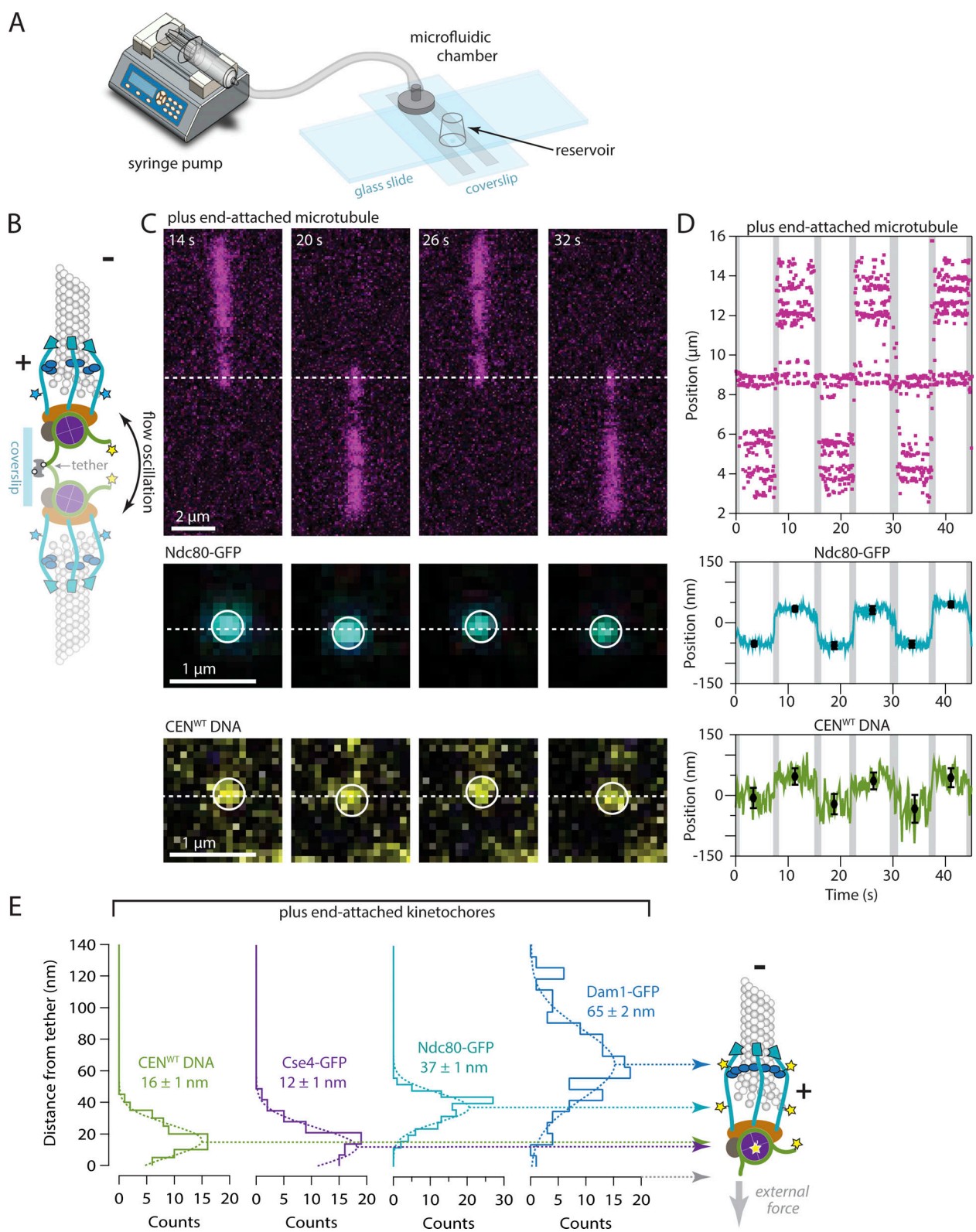

**Figure 3. Plus-end-attached kinetochores are well organized along the microtubule axis. (A)** Kinetochores were assembled in a microfluidic device and then allowed to capture microtubules. A syringe pump enabled imaging of the kinetochores and their captured microtubules while the buffer flowed gently through the assembly chamber. **(B)** Schematic of a surface-assembled kinetochore attached to the tip of a microtubule. Oscillating the direction of flow caused the kinetochore and its captured microtubule to flip back and forth, reorienting by 180° around the biotin-avidin tether with each reversal of the flow. **(C)** Time-lapse image series showing flow-induced reorientation of a microtubule (magenta) attached by its end to a surface-assembled kinetochore. Both the Ndc80-GFP kinetochore marker (cyan) and the Atto565 label on the wild type centromeric DNA (yellow, CEN^WT) oscillated with the direction of buffer flow. Horizontal dashed lines indicate approximate positions of the DNA tether point on the coverslip. **(D)** Example records of position versus time for an Ndc80-GFP spot and

the corresponding Atto565-labeled centromeric DNA obtained by tracking the individual spots with subpixel accuracy. Displacements of each spot from the tether point were estimated by averaging during the intervals when the microtubule orientation was steady. The position of the biotin–avidin tether point was inferred as the midpoint between tracked positions before and after each flow reversal. Black symbols represent mean ± SD from $N = 60$ tracked positions during each interval. Positions recorded during the reorientation of the microtubule were omitted from the averaging and are indicated here by gray shading. Additional records are shown in Fig. S4. **(E)** Distributions of displacement for the indicated fluorescent kinetochore components (from $N = 67$–128 intervals), fit with single Gaussian functions. The mean ± SEM for each Gaussian is indicated. Displacements for Cse4-GFP, a component of the centromeric nucleosome, are similar to the centromeric DNA (CEN$^{WT}$), as expected. The larger displacements for outer microtubule-binding components, Ndc80-GFP and Dam1-GFP, are consistent with the in vivo arrangement (Joglekar et al., 2009; Cieslinski et al., 2023).

---

early in mitosis. A variety of mechanisms have been proposed to explain the astounding fidelity of mitosis, but they have focused almost exclusively on plus-end attachments (Funabiki, 2019; Lampson and Grishchuk, 2017; Sarangapani and Asbury, 2014). Our findings now indicate that the discrimination between correct and incorrect attachments can begin even before both sisters have achieved plus-end attachments. In prometaphase, sister kinetochores are exposed to many spindle microtubules emanating from both poles. After a pair of sister kinetochores initially makes side attachments, one of them will (by chance) become tip-attached before the other, tracking with tip shortening and exerting elastic pulling forces on its side-attached sister. If the pair is attached incorrectly to microtubules from the same pole, then the side-attached sister will be pulled toward the minus end. Its grip will therefore be weak (Fig. 5 A, left), and it will likely detach. Conversely, if the pair is attached correctly to microtubules from opposite poles, then the side-attached sister will be pulled toward the plus end. It will therefore have a stronger grip (Fig. 5 A, right) that should allow it to remain attached, increasing the likelihood that it will achieve proper biorientation, either by sliding all the way to the plus end or by capturing the plus end during disassembly. The greater frictional resistance of the correctly side-attached sister will also cause the end-attached kinetochore to experience higher force, stabilizing its end-attachment by the catch bond-like effect we previously discovered (Akiyoshi et al., 2010) and protecting it from Aurora B kinase–triggered detachment (de Regt et al., 2022). Thus the asymmetric grip of the side-attached sister can selectively stabilize arrangements that are on the pathway toward plus-end biorientation in several ways.

The mechanical asymmetry of the kinetochore appears to arise at least partly from its primary microtubule-binding subcomplex, Ndc80c. The stalk of Ndc80c emerges from the "foot" (from the calponin-homology domains) with a tilt toward the microtubule plus end (Cheeseman et al., 2006; Alushin et al., 2010; Muir et al., 2023). In principle, this local structural asymmetry by itself could cause asymmetric mechanical behavior, such that when a side-attached kinetochore is pulled toward the plus end, its Ndc80c feet may bind more strongly. When pulled toward the minus end, its feet may bind more weakly. Given that the yeast Ndc80c alone is relatively weak compared to native yeast kinetochore particles, and that even the human Ndc80 complex showed less dramatic mechanical asymmetry than the yeast kinetochore particles, additional subcomplexes probably contribute to the kinetochore's direction-dependent grip. After emerging asymmetrically from the foot, the Ndc80c stalk contains a flexible "hinge" (Wang et al., 2008; Zahm et al., 2023; Polley et al., 2023), which suggests that the

rest of the stalk should align at least partially with the direction of external force. Indeed, our data imply that pulling a kinetochore toward the plus end aligns its Ndc80c stalks into a parallel configuration, potentially facilitating interactions with Dam1c oligomers that can provide additional microtubule contacts and strengthen the overall grip of the kinetochore on the microtubule (Fig. 5 B) (Tien et al., 2010; Umbreit et al., 2014; Lampert et al., 2010). At a plus end, the Ndc80c stalks can project past the tip of the microtubule to converge onto the centromeric nucleosome, potentially allowing Dam1c oligomers to organize a cage-like arrangement surrounding the tip (Jenni and Harrison, 2018; Muir et al., 2023) that further increases grip strength (Fig. 5 C). Pulling a kinetochore toward the minus end disrupts the parallel organization of Ndc80c stalks, likely preventing Dam1c from making additional microtubule contacts, weakening the kinetochore's grip when side-attached, and also precluding strong attachment to the minus end. An analogous mechanism could occur in humans (which lack Dam1c) if the parallel plus-end-directed alignment of neighboring human Ndc80 complexes promotes their clustering, which significantly strengthens their binding to microtubules and can occur via a short loop domain within the stalk (Polley et al., 2023).

Our isolated kinetochores captured microtubules with a strong preference for plus ends, and their attachment strength was higher at plus ends than at minus ends. A similar observation was made previously using chromosomes isolated from CHO cells (Huitorel and Kirschner, 1988). Because we imaged the captured microtubules only after washing out excess unbound filaments, we did not directly observe the capture process. However, a two-step process seems likely, where both plus and minus ends bind initially, but the mechanically weaker minus end attachments are preferentially lost due to viscous forces during the washout. We can exclude a role for active motors because our kinetochores lacked motors (Akiyoshi et al., 2010; Lang et al., 2018) and ATP was absent from our experiments. Our capture assays used filaments stabilized by GMPcPP and taxol, indicating that preferential plus-end capture does not require GTP caps or microtubule dynamics, and probably does not require curved tubulin since the ends of GMPcPP microtubules are usually blunt (Wieczorek et al., 2015). Moreover, we found that side-attached kinetochores were sensitive to microtubule polarity even far away from the ends of the filaments. We propose that all three of these intrinsic kinetochore behaviors—their preference for capturing (i) and holding microtubule plus ends under tension (ii), and their directionally asymmetric grip when side-attached (iii)—could all arise from the structural polarity of the microtubule and how it influences kinetochore architecture (Fig. 5, B and C).

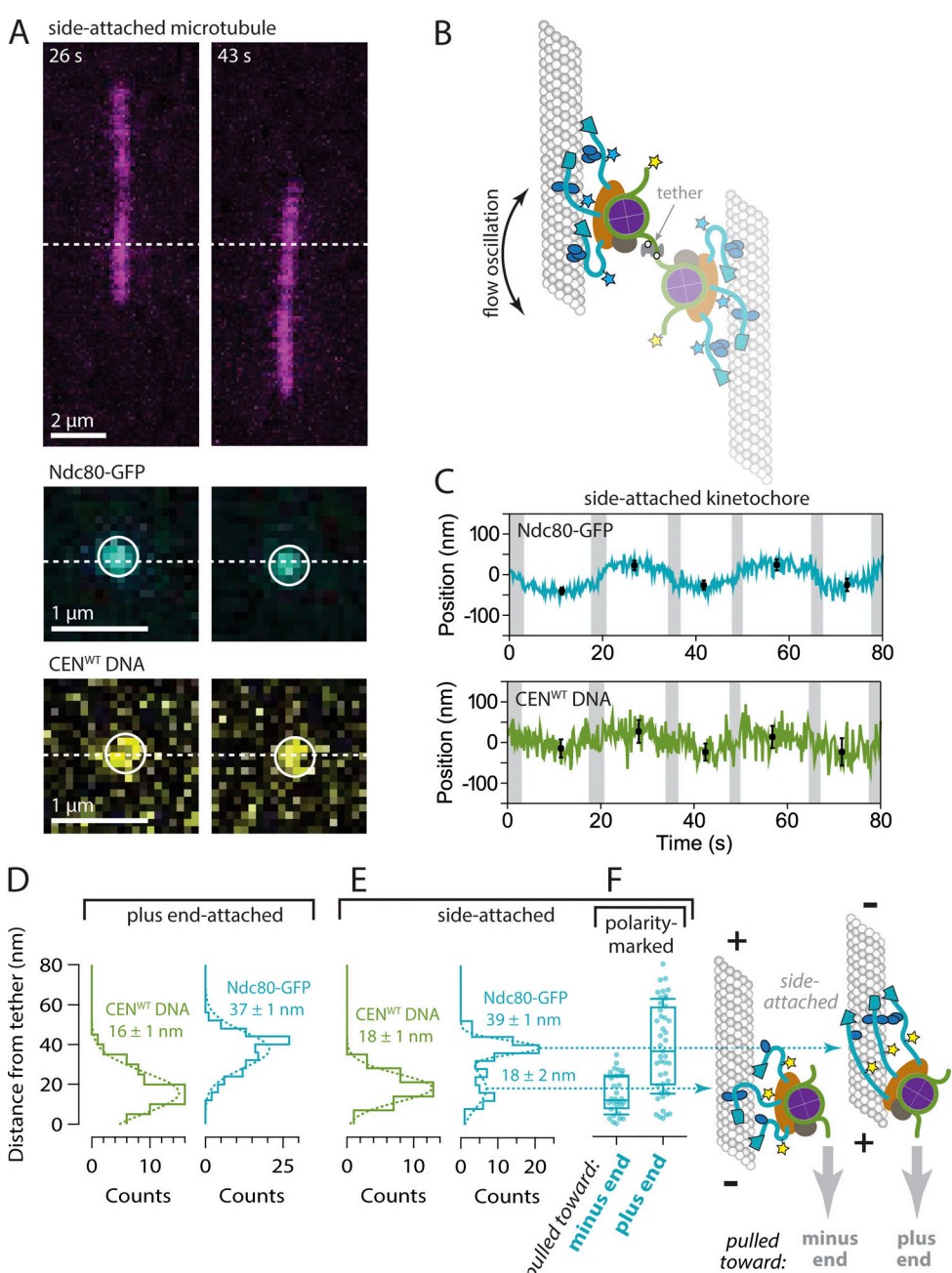

Figure 4. **Side-attached kinetochores are more compact specifically when pulled toward minus ends. (A)** Time-lapse images showing flow-induced reorientation of a microtubule (magenta) attached by its side to a surface-assembled kinetochore. Both the Ndc80-GFP kinetochore marker (cyan) and the Atto565 label on the centromeric DNA (yellow) oscillated with the direction of buffer flow. Horizontal dashed lines indicate approximate positions of the DNA tether point on the coverslip. **(B)** Schematic of a surface-assembled kinetochore attached to the side of a microtubule. Oscillating the direction of flow caused the kinetochore and its captured microtubule to flip back and forth, reorienting by 180° with each reversal of the flow. **(C)** Example records of position versus time for an Ndc80-GFP kinetochore attached to the side of a microtubule. Displacements of both the Ndc80-GFP spot and the Atto565 label on the wild type centromeric DNA (CEN$^{WT}$) relative to the tether point were estimated by averaging during the intervals when the microtubule orientation was steady. Black symbols represent mean ± SD from N = 60 tracked positions during each interval. Positions recorded during the reorientation of the microtubule were omitted from the averaging and are indicated here by gray shading. **(D)** Distributions of displacement for the indicated fluorescent components within tip-attached kinetochores (from N = 74–116 intervals), fit with single Gaussian functions. The mean ± SEM for each Gaussian is indicated. These data and fits are replotted from Fig. 3 E with an expanded vertical scale. **(E)** Distributions of displacement for the indicated fluorescent components within side-attached kinetochores (from N = 32–87 intervals), fit with either a single Gaussian (CEN$^{WT}$ DNA) or a double Gaussian function (Ndc80-GFP). The mean ± SEM for each Gaussian is indicated. The distribution of Ndc80-GFP displacements for side-attached kinetochores is wider in comparison to the tip-attached kinetochores in panel D, and apparently bimodal, including a sub-population very close to the tether with a mean displacement of only 18 ± 2 nm. **(F)** Distributions of displacement for Ndc80-GFP within kinetochores attached to the sides of polarity-marked microtubules. Boxes extend from first to third quartiles with medians indicated by central horizontal solid lines. Medians for kinetochores pulled toward plus and minus ends were 36 and 12 nm, respectively (from N = 45 and 30 intervals). Whiskers extend ± one SD from the mean.

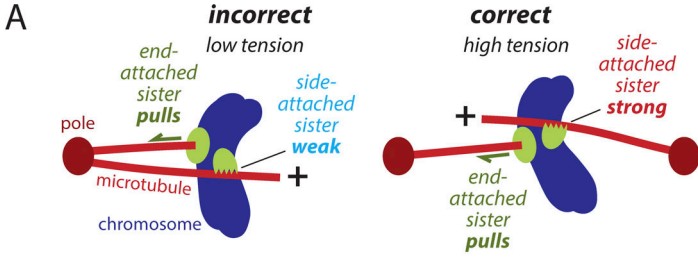

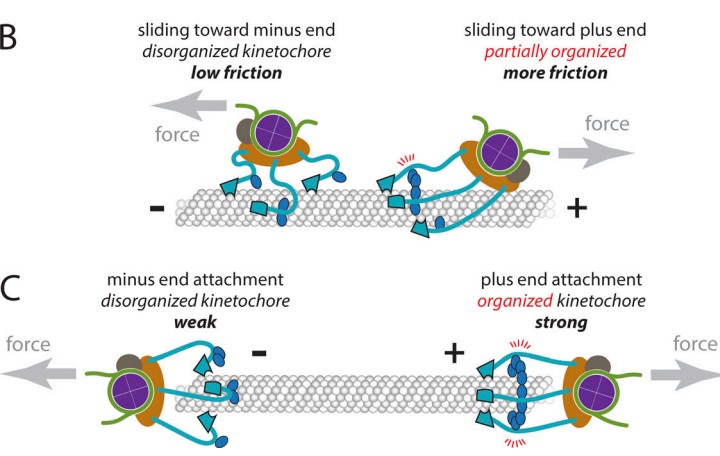

Figure 5. **How the directionally asymmetric grip of the kinetochore can promote accuracy during early mitosis and how it might arise from microtubule polarity. (A)** After a pair of sister kinetochores initially makes side-attachments, one of them will (by chance) become tip-attached before the other, tracking with tip shortening and exerting elastic pulling forces on its side-attached sister. Left: If the pair is attached incorrectly to microtubules from the same pole, then the side-attached sister will be pulled toward the minus end. Its grip will therefore be weak and it will likely detach. Right: If the pair is attached correctly to microtubules from opposite poles, then the side-attached sister will be pulled toward the plus end. It will therefore have a stronger grip that should allow it to remain attached and achieve proper biorientation at the plus end. **(B)** Each Ndc80c fibril (light blue) has a globular foot (outlined in black), which binds with a stereospecific "footprint" on the outside surface of the microtubule, and a coiled-coil stalk that emanates from this foot and projects toward the plus end. Right: Pulling a kinetochore toward the plus end aligns its multiple Ndc80c stalks into a parallel configuration, facilitating interactions with Dam1c oligomers and strengthening the overall grip on the microtubule. Because the direction of force is aligned with the stalks, torque on the Ndc80c-microtubule bonds is minimized. Left: Pulling a kinetochore toward the minus end disrupts this organization, weakening its grip. **(C)** Right: At a plus end, the Ndc80c stalks can all project in parallel past the tip of the microtubule and converge onto the centromeric nucleosome, potentially allowing Dam1c oligomers to organize a cage-like arrangement surrounding the tip (Jenni and Harrison, 2018; Muir et al., 2023) with even higher grip strength. Left: At a minus end, parallel convergence of the stalks is impossible, increasing the torque on the Ndc80c-microtubule bonds and potentially reducing lateral interactions via Dam1c.

Recent studies have revealed directionally asymmetric gripping of F-actin by a number of focal adhesion proteins (Huang et al., 2017; Owen et al., 2022; Arbore et al., 2022). This behavior is thought to drive the self-assembly of organized focal adhesions with appropriately oriented F-actin filaments, and therefore to underlie cellular sensing of directional physical cues (Swaminathan et al., 2017). The kinetochore's asymmetric grip probably serves an analogous role, selectively stabilizing attachments to correctly oriented microtubules at the chromosome-spindle junction during mitosis. Our work suggests that asymmetric gripping may be a general phenomenon underlying the self-assembly of cytoskeletal junctions with productively oriented cytoskeletal filaments in many cellular contexts.

## Materials and methods
Reagents and resources are listed in Table 1.

### Yeast strain construction
All strains described in this study are derivatives of SBY3 (W303). Generation of *Saccharomyces cerevisiae* strains harboring GFP-tagged kinetochore proteins and a phospho-mimetic mutation in Dsn1 (Dsn1-2D) that has been shown to enhance outer kinetochore assembly (Akiyoshi et al., 2013; Lang et al., 2018) was achieved either by standard genetic crosses and media selection (Amberg et al., 2005) or using standard PCR-based integration at the endogenous loci (Longtine et al., 1998). All yeast transformants were confirmed by PCR or sequencing.

### Preparation of yeast whole-cell lysates
Yeast whole-cell lysates used for de novo kinetochore assembly were prepared essentially as described in prior publications (Lang et al., 2018; Popchock et al., 2023) and summarized as follows: Cells were grown in 2 liters of liquid yeast peptone dextrose (YPD) media at room temperature and harvested in log phase by centrifugation. Cell pellets were placed on ice and washed with ice-cold milli-Q purified water plus 0.2 mM PMSF and centrifuged. Pellets were washed a second time with ice-cold Buffer L (25 mM HEPES pH 7.6, 2 mM $MgCl_2$, 0.1 mM EDTA, 0.5 mM EGTA, 0.1% NP-40, 175 mM K-Glutamate, and 15% Glycerol) plus 2 mM DTT and protease inhibitors (10 μg/ml leupeptin, 10 μg/ml pepstatin, 10 μg/ml chymostatin, and 10 μM PMSF) and centrifuged again. Cells were resuspended in a final volume of Buffer L given by $u = v\,o$, where $u$ represents the volume in μl of added Buffer L, $v$ represents the original volume in ml of the liquid culture, and $o$ represents the optical density of the culture measured at the time of harvest. The cellular resuspension was snap-frozen as small spherical pellets by pipetting drops of the suspension directly into liquid nitrogen. Cell lysis was achieved using a Freezer/Mill (SPEX SamplePrep) by alternating milling of the pellets at 10 Hz for 2 min followed by a 2-min cooling phase for 10 cycles. The resulting lysate powder was then thawed on ice and clarified by centrifugation at 16,100 $g$ for 30 min at 4°C. The protein-containing supernatant was subsequently aliquoted (100 μl) and snap-frozen in liquid nitrogen. Aliquots were stored at −80°C until use.

Table 1.  **Reagents and resources**

| Reagent or resource | Source | Identifier |
| --- | --- | --- |
| **Antibodies** | | |
| His Tag biotinylated antibody | R&D Systems | BAM050 |
| **Biological samples** | | |
| Purified tubulin from bovine brains | Schenk Packing Co. | NA |
| Yeast whole-cell lysate | Listed strains | NA |
| **Chemicals, peptides, and recombinant proteins** | | |
| Tubulin protein (fluorescent HiLyte 647): Porcine brain | Cytoskeleton, Inc. | TL670M |
| Tubulin protein (biotin): Porcine brain | Cytoskeleton, Inc, | T333P |
| Vectabond reagent, tissue section adhesion | Vector Laboratories | SP-1800-7 |
| mPEG-SVA MW 5,000 | Laysan Bio, Inc. | mPEG-succinimidyl valerate |
| Biotinylated mPEG-SVA MW 5,000 | Laysan Bio, Inc. | Biotin-PEG-SVA |
| Avidin DN | Vector Laboratories | A-3100-1 |
| Biotin-BSA | Vector Laboratories | B-2007-10 |
| Paclitaxel | Millipore Sigma | T7191-5MG |
| Guanosine 5'-triphosphate sodium salt hydrate | Milipore Sigma | G8877-1G |
| Streptavidin-coated polystyrene beads | Spherotech | SVP-05-10 |
| Guanosine-5'-[(α,β)-methyleno] triphosphate, sodium salt (GMPcPP) | Jena Bioscience | JBS-NU-405S |
| **Experimental models: Organisms/strains** | | |
| *S. cerevisiae*: MATa ura3-1::pCSE4-CSE4-XbaI(GFP):URA3 leu2,3-112 his3-11 trp1-1 ade2-1 LYS2+ can1-100 bar1 cse4Δ::KanMX | Biggins lab | SBY19926 |
| *S. cerevisiae*: MATa pDsn1-Dsn1-2D-3FLAG:URA3 Ndc10-GFP:kanMX6 | Biggins lab | SBY21618 |
| *S. cerevisiae*: MATa pDsn1-Dsn1-2D-3FLAG:URA3 Ndc80-GFP:kanMX6 | Biggins lab | SBY21620 |
| *S. cerevisiae*: MATa pDsn1-Dsn1-2D-3FLAG:URA3 Ctf19-GFP:kanMX6 | Biggins lab | SBY21621 |
| *S. cerevisiae*: MATx pDsn1-Dsn1-2D-3FLAG:URA3 Dam1-3GFP:HIS | Biggins lab | SBY20634 |
| **Oligonucleotides** | | |
| Forward primer for generating 208-bp *CEN3* or *CEN3*[mut]: /5BiosG/5'-GGTGGTTCTGGTGGTTCTGGTGAATTC-CCATTCAATGAAATATATATTTCTTACTATTTC-3' | This paper | 50_JDL |

Table 1. **Reagents and resources (Continued)**

| Reagent or resource | Source | Identifier |
|---|---|---|
| Reverse primer for generating 208-bp CEN3 or CEN3<sup>mut</sup>: /5Atto565N/5′-GCTATTCAT TGAAAAAATAGTACAAATAAG-3′ | This paper | 52_JDL |
| 5′ primer to tag Ndc80: 5′-ACG AAATTTGGAGTTTGAAACTGA ACATAACGTAACAAATCGGAT CCCCGGGTTAATTAA-3′ | This paper | 30_JDL |
| 3′ primer to tag Ndc80: 5′-CTG TAGATTGCTCGGGTATTATAT ATCATTTATTTTATTAGAATT CGAGCTCGTTTAAAC-3′ | This paper | 31_JDL |
| 5′ primer to tag Ctf19: 5′-GAT CTGCAACGTTTGCCTATTCCC GGACATGTACGCCAGGCGGAT CCCCGGGTTAATTAA-3′ | This paper | 10_JDL |
| 3′ primer to tag Ctf19: 5′-TAA GCAAGCCGTCCAGTTGGCAAT GGCAAATGGAACATCAGAATT CGAGCTCGTTTAAAC-3′ | This paper | 11_JDL |
| 5′ primer to tag Ndc10: 5′-TCA AAATTCATTTGATGGTCTGTT AGTATATCTATCTAACCGGAT CCCCGGGTTAATTAA-3′ | This paper | 6_JDL |
| 3′ primer to tag Ndc10: 5′-TAT CCCTATACGAAACAGTTTAAA CTTCGAAGCTCCCTCAGAATT CGAGCTCGTTTAAAC-3′ | This paper | 7_JDL |
| **Recombinant DNA** | | |
| pSB963: WT CEN3, 8LacO, TRP1 | Lang et al. (2018) | NA |
| pSB972: Mutant CEN3 (CCG->AGC CEN mutant), 8 LacO, TRP1 | Lang et al. (2018) | NA |
| **Software and algorithms** | | |
| Single molecule colocalization analysis (written in LabView) | This paper | https://github.com/casbury69/ smTIRF-spot-selection- colocalization-and-brightness- vs-time |
| Displacement analysis (written in Igor Pro) | This paper | https://github.com/casbury69/ flip-flop-IGOR-analysis-routines |
| ImageJ (Fiji) | Schindelin et al. (2012) | https://imagej.net/software/ fiji/ |
| MOSAICsuite | Sbalzarini and Koumoutsakos (2005) | https://imagej.net/plugins/ mosaicsuite |

### Preparation of centromeric DNAs

208-bp Atto565-labeled wild type and mutant centromeric DNAs were generated by PCR from plasmids pSB963 and pSB972, respectively, which are both based on the centromeric DNA sequence from *S. cerevisiae* chromosome III (CEN3). The mutant centromeric DNAs contained a 3-bp substitution in the CDEIII region that blocks kinetochore assembly in vivo (Lechner and Carbon, 1991; Sorger et al., 1994, 1995) and in vitro (Lang et al., 2018). The forward 5′ primer contained a 5′ biotin for specific attachment to the coverslip and the reverse primer was labeled

with Atto565. Both primers were custom-synthesized, including the biotin and Atto565 labels, by Integrated DNA Technologies. PCR products were purified using a Qiagen PCR Cleanup kit and eluted into milli-Q purified water.

### Expression and purification of recombinant Ndc80 complexes

Yeast and human Ndc80 complexes were expressed and purified essentially as described in prior publications (Hamilton et al., 2020; Helgeson et al., 2018) and summarized as follows: Proteins were co-expressed in BL21(DE3) Rosetta 2 *E. coli* cells

(Stratagene) from two bicistronic plasmids, one encoding Spc25 and His$_6$-tagged Spc24, and another encoding either human Hec1 and Nuf2 or yeast Ndc80 and Nuf2, in the pST39 backbone. Cells were lysed with a French press and the lysate was clarified by centrifugation. Ndc80 complexes were purified using nickel-charged nitrilotriacetic acid (Ni-NTA) affinity chromatography and buffers supplemented with protease inhibitors. Affinity chromatography was followed by size-exclusion chromatography using a Superdex 200 16/60 column. A bicinchoninic acid (BCA) assay was used to determine the Ndc80 complex concentration. Coomassie-stained SDS-PAGE analyses of the recombinantly purified Ndc80 complexes are provided in Fig. S5.

## Purification of native kinetochore particles

Native kinetochore particles were purified from asynchronously grown *S. cerevisiae* SBY8253 cells (Genotype: MATa DSN1-6His-3Flag:URA3) grown in YPD medium (2% glucose) by modifying previous protocols (Akiyoshi et al., 2010; Miller et al., 2016). Protein lysates were prepared using a freezer mill (SPEX SamplePrep) submerged in liquid nitrogen. Lysed cells were resuspended in buffer H (25 mM Hepes, pH 8.0, 2 mM MgCl$_2$, 0.1 mM EDTA, 0.5 mM EGTA, 0.1% NP-40, 15% glycerol, and 150 mM KCl) containing phosphatase inhibitors (0.1 mM Na-orthovanadate, 0.2 µM microcystin, 2 mM β-glycerophosphate, 1 mM Na pyrophosphate, and 5 mM NaF) and protease inhibitors (20 µg/ml leupeptin, 20 µg/ml pepstatin A, 20 µg/ml chymostatin, and 200 µM PMSF). Lysates were ultracentrifuged at 98,500 $g$ for 90 min at 4°C. Dynabeads (catalog number 112-05D; Invitrogen) were conjugated with an anti-FLAG antibody (catalog number F3165; Sigma-Aldrich), and immunoprecipitation of Dsn1-6His-3Flag was performed at 4°C for 3 h. After immunoprecipitation, the Dynabeads were washed once with lysis buffer containing 2 mM DTT and protease inhibitors, three times with lysis buffer with protease inhibitors, and once in lysis buffer without inhibitors. Kinetochore particles were then eluted by gentle agitation of beads in elution buffer (buffer H plus 0.5 mg/ml 3FLAG Peptide, which was custom synthesized by Lot# 7765380001/PE6749; GenScript) for 30 min at room temperature.

## Preparation of taxol-stabilized microtubules

Purified bovine brain tubulin was added to microtubule polymerization buffer (1× BRB80 [80 mM PIPES, 1 mM MgCl$_2$, 1 mM EGTA], 7% DMSO, 4 mM MgCl$_2$, and 1 mM GTP) to a final concentration of 2 mg/ml and incubated at 37°C for 1 h. After 1 h, 3 µl of prewarmed 1× BRB80 + 10 µM taxol was added for every 1 µl of polymerized microtubules. Taxol-stabilized microtubules were then spun for 10 min at 90,000 RPM (TLA100.0; Beckman Optima MAX-XP) at 37°C. The microtubule pellet was resuspended in 150 µl of 1× BRB80 + 10 µM taxol and stored at room temperature. To generate fluorescent or biotinylated microtubules, porcine HyLite 647 or biotin tubulin (Cytoskeleton) was added (6% wt/wt) to the polymerization reaction.

## Slide passivation for single-molecule TIRF microscopy

Slides were prepared essentially as described by Crawford et al. (2008) and summarized as follows: coverslips and slides were plasma cleaned for 10 min followed by four sequential hour-long sonications in 2% Micro-90, 200 proof ethanol, 1 M KOH, and finally Milli-Q water. Slides and coverslips were then completely dried using ultrapure nitrogen. After drying, slides and coverslips were treated with Vectabond (Vector Laboratories) dissolved in acetone (1% vol/vol) for 5–10 min. Slide chambers were constructed by sandwiching the coverslip and slide together with double-sided tape. Passivation was achieved by adding a 1:100 Biotinylated mPEG-SVA/mPEG-SVA MW 5,000 in 0.1 M sodium bicarbonate (1% wt/vol). Chambers were incubated with polyethylene glycol (PEG) overnight at room temperature.

## Kinetochore assembly in whole-cell extracts and microtubule capture assays

Excess mPEG solution was washed out with 400 µl of 1× BRB80 and then blocked with a 0.1 mg/ml BSA solution for 5 min. The chamber was then washed with 200 µl of 1× BRB80. After blocking, a solution of 0.3 mg/ml avidin was added to the chamber for 5 min and washed with an additional 200 µl of 1× BRB80. Following the addition of avidin, 50–200 pM biotinylated Atto565 CEN DNAs were introduced into the chamber and incubated for 5 min. Excess DNA was washed away with 200 µl of 1× BRB80. To assemble surface-tethered kinetochores, 100 µl of yeast whole-cell lysate was added to chambers with surface-tethered CEN DNAs and incubated for 1 h. For colocalization assays, the lysate was washed away with 400 µl of 1× BRB80 with glucose oxidase (165 U/ml), catalase (217 U/ml), and 0.65% glucose (wt/vol) for scavenging oxygen. For microtubule capture assays, taxol-stabilized microtubules were diluted 1:3 in BRB80 + 10 µM taxol and briefly sheared using a vortexer for 25 s before introduction into the slide chamber. After a 15-min incubation, excess microtubules were washed away with 400 µl of 1× BRB80 with glucose oxidase/catalase. All slide preparation, kinetochore assembly, microtubule capture, and imaging were performed at 21°C.

## Single-molecule colocalization analysis

All images were collected on a custom TIRF microscope built on a standard Nikon TE inverted microscope base and using a Nikon Apo TIRF 100× 1.49 NA oil-immersion objective lens (Deng and Asbury, 2017). Excitation of fluorescent proteins and organic dyes was achieved using expanded beams from three solid-state lasers at 488 nm (Coherent Sapphire), 561 nm (Coherent Sapphire), and 641 nm (Coherent Cube). Images were acquired with three separate Andor iXon897+ EMCCD cameras. For colocalization assays, 20–60 frames were collected with 0.5 s integrations. Analysis was performed using custom Labview (National Instruments) software available at https://github.com/casbury69/smTIRF-spot-selection-colocalization-and-brightness-vs-time. The software implements spot-picking for each fluorescent channel using methods described by Crocker and Grier (1996), and by Friedman and Gelles (2015). Mapping between color channels was performed by creating a linear registration map using blue/green/orange/dark red 500 nm beads (T7281; Tetraspec) as fiducials.

## Preparation of polarity-marked microtubules

To produce polarity-marked microtubules, two seed mixes were prepared on ice: a bright seed mix (13.3 µM unlabeled bovine

tubulin, 6.7 µM Hilyte 647 tubulin, 1 mM DTT, 1 mM GMPcPP, 1× BRB80) and a dim seed mix (9 µM unlabeled bovine tubulin, 1 µM HyLite 647 cytoskeletal tubulin, 8 µM N-ethylmaleimide [NEM]–treated bovine tubulin, 1 mM DTT, 1 mM GMPcPP, 1× BRB80). Each seed mix was clarified using an ultracentrifuge (90,000 RPM, 4°C, 5 min; Beckman Optima MAX-XP) and then snap-frozen in small aliquots and stored at –80°C. Bright seeds were polymerized by diluting an aliquot of bright seed mix fivefold (vol/vol) in 1× BRB80 with 1 mM DTT. The diluted bright seed mix was then incubated for 45 min at 37°C to allow for polymerization. To grow dim elongations from the bright seeds, an aliquot of dim seed mix was diluted 5.7-fold (vol/vol) in 1× BRB80 with 1 mM DTT on ice and then warmed for 20 s at 37°C. Polymerized bright seeds were added 4.4-fold (vol/vol) to the dim mix and incubated for 1 h at 37°C. After the second polymerization with the dim seed mix, the microtubules were centrifuged for 5 min at 22,000 RPM (TLA100.0; Beckman Optima MAX-XP) at 37°C. The pellet was then resuspended in 150 µl assembly assay buffer (1× BRB80, 1 mM DTT, 0.025 mg/ml K-casein, 20 µM taxol).

### Tracking kinetochore subunit displacements and estimating intrakinetochore distances

Custom flow chambers, with an attached reservoir to hold excess buffer and with custom-made fittings, were designed to generate a gentle oscillating flow to orient surface-assembled kinetochore/microtubule pairs along the coverslip surface. The fitting was attached to a syringe pump (780210; Kd Scientific) which operated at a slow flow rate of 0.6 ml/min. The volume of each oscillation was 0.2 ml. Images were acquired in the same manner used during the colocalization assays with the exception of using 0.2 s integrations instead of 0.5 s integrations. The total number of frames collected was determined by the bleach rate of GFP for the individual kinetochores. Particle tracking was performed using the MOSAICsuite 2D particle tracker plugin for ImageJ (Sbalzarini and Koumoutsakos, 2005). Analysis of particle displacements was achieved using custom scripts written in IgorPro (Wavemetrics) and available at https://github.com/casbury69/flip-flop-IGOR-analysis-routines.

### Measuring the rupture strength of end-attached kinetochore particles

Rupture force assays were carried out as described in prior publications (Akiyoshi et al., 2010; Miller et al., 2016) and summarized as follows: dynamic microtubules were grown from biotinylated-GMPcPP seeds anchored on coverslips passivated with biotinylated BSA in microtubule growth buffer (BRB80, 1 mM GTP, 250 µg/ml glucose oxidase, 25 mM glucose, 30 µg/ml catalase, 1 mM DTT, 24 µM purified bovine brain tubulin, and 0.5 mg/ml κ-casein). Polystyrene beads coated with anti-HIS antibody (BAM050; R&D systems) were prepared and stored as previously described.(Sarangapani et al., 2021) Immediately before each experiment, 6 pM anti-His beads were incubated for 15 min at 4°C with purified kinetochore material, corresponding to Dsn1-His-Flag concentrations ranging between 2 and 4 nM. Kinetochore-decorated beads were then diluted 8- to 10-fold in a solution of growth buffer containing 1.5 mg ml⁻¹ purified bovine

brain tubulin and an oxygen scavenging system (1 mM DTT, 500 µg ml⁻¹ glucose oxidase, 60 µg ml⁻¹ catalase, and 25 mM glucose) and then introduced into the slide chamber.

For efficiency of data collection, beads that were already bound to microtubules (on the lattice, away from the dynamic tip) were usually chosen for measurements of rupture strength. Initially, the attachments were preloaded with a constant tensile force of 1–3 pN, which caused the lattice-bound beads to slide until reaching the microtubule end. Once they were at the end, we verified that the beads moved under the preload force at a rate consistent with that of microtubule growth. The laser trap was subsequently programmed to ramp the force at a constant rate (0.25 pN s⁻¹) until the linkage ruptured or the load limit of the trap was reached (~23 pN under the conditions used here). Fewer than 5% of all trials ended in detachment during the preload period before force ramping began, while 0–15% reached the load limit. We also tested beads that were floating freely in solution to estimate the fraction of active beads that were capable of binding microtubules, which remained <50%, thus ensuring single-particle conditions (Akiyoshi et al., 2010; Sarangapani et al., 2013). Bead position was recorded using custom LabView software and analyzed to determine the rupture force with custom scripts in IgorPro. Plus and minus ends were distinguishable because the plus ends grew faster, extending farther from the coverslip-anchored, GMPcPP seeds than the minus ends.

### Measurement of directional sliding friction

Sliding friction measurements were performed either using beads decorated sparsely with native kinetochore particles, which were prepared as described above, or using beads densely coated with recombinant yeast or human Ndc80c, which were prepared as follows. Immediately before each experiment, 6 pM anti-His beads were incubated for 60 min at 4°C with purified 10 nM Ndc80c such that each bead was decorated with ~3,000 protein complexes. The beads were then washed by pelleting and resuspension in growth buffer, diluted 8- to 10-fold into growth buffer containing 1.5 mg ml⁻¹ purified bovine brain tubulin and our oxygen scavenging system (detailed above), and then introduced into the slide chamber. Based on simple geometric considerations (detailed in Hamilton et al. [2020]), we estimate that a maximum of ~90 Ndc80 complexes on each bead would be capable of simultaneously binding the microtubule surface under the conditions we tested.

For all the sliding friction measurements, microtubules were polymerized as described above with the addition of a 10-min incubation with a growth buffer containing a higher concentration (24 µM) of tubulin to allow minus-end extensions to grow long enough for bead binding and sliding along their sides. The laser trap was then used to apply a constant force in one direction along the longitudinal axis of the microtubule until the bead slid ~1 µm. The direction of applied force was then reversed until ~1 µm in the opposite direction. This was repeated several times for each bead–microtubule pair, at forces varying between 0.5 and 4 pN on both minus- and plus-end extensions of the microtubules. Sliding velocities were measured using linear regression in IgorPro.

## Online supplemental material

The manuscript includes five supplemental figures, one supplemental Excel spreadsheet file, and one source data file. The contents of each of these supplemental materials is summarized briefly below. Fig. S1 shows de novo assembly of individual kinetochores occurs specifically on centromeric DNAs. Fig. S2 shows plus-end preference is not an artifact of differential labeling. Fig. S3 shows example records showing measurement of bidirectional sliding friction. Fig. S4 show nanoscale displacement of fluorescent-tagged components within individual plus-end-attached kinetochore assemblies during periodic flow-induced reorientation. Fig. S5 shows Coomassie-stained SDS-PAGE analyses of the recombinantly purified yeast and human Ndc80 complexes. Video 1 shows time-lapse TIRF microscopy of assembled kinetochore with GFP-tagged Ndc80 (cyan) bound to the tip of an Alexa647-labeled taxol-stabilized microtubule (magenta) from Fig. 1 D. Video 2 shows time-lapse TIRF microscopy of assembled kinetochore with GFP-tagged Ndc80 (cyan) bound to the side of an Alexa647-labeled taxol-stabilized microtubule (magenta) from Fig. 1 D. Video 3 shows time-lapse TIRF microscopy of assembled kinetochore with a GFP-tagged Ndc80 (cyan) bound to the tip of an Alexa647-labeled taxol-stabilized microtubule (magenta) with flow-induced oscillation (left) from Fig. 3 C. Video 4 shows time-lapse TIRF microscopy of assembled kinetochore with a GFP-tagged Ndc80 (cyan) bound to the side of an Alexa647-labeled taxol-stabilized microtubule (magenta) with flow-induced oscillation (left) from Fig. 4 A. Table S1 shows all the individual measured values used to create the graphs in the paper. Table S2 lists primers, plasmids, and strains.

## Data availability

The data underlying Fig. 1 C; Fig. S1; Fig. 2, C, F, and I; Fig. 3 E; and Fig. 4, E and F are available in the online supplemental material, in an Excel spreadsheet file titled Table S1.

## Acknowledgments

We thank Trisha Davis, Bonnibelle Leeds, Lucia Kyung-Ae Maki-Fern, and Juan-Jesus Vicente for critical reading and feedback on the manuscript, and all the members of the Davis, Asbury, and Biggins labs for insightful discussions. We also thank Christian Nelson (Basic Sciences Division, Fred Hutchinson Cancer Research Center, Seattle, WA, USA) and Cameron Lee (Basic Sciences Division, Fred Hutchinson Cancer Research Center, Seattle, WA, USA) for constructing strains and purifying kinetochores.

This project was funded by the National Institutes of Health (T32HL007312 to J.D. Larson, F32GM136010 to A.R. Popchock, R01GM079373 and R35GM134842 to C.L. Asbury, and R01GM064386 to S. Biggins), a Washington Research Foundation Postdoctoral Fellowship to J.D. Larson, and a Mary Gates Research Scholarship to N.A. Heitkamp (Mary Gates Endowment for Students, University of Washington, Seattle, WA, USA). S. Biggins is an investigator of the Howard Hughes Medical Institute.

Author contributions: J.D. Larson: Conceptualization, Data curation, Formal analysis, Funding acquisition, Investigation, Methodology, Project administration, Software, Supervision, Validation, Visualization, Writing - original draft, Writing - review & editing, N.A. Heitkamp: Conceptualization, Data curation, Formal analysis, Funding acquisition, Investigation, Methodology, Validation, Visualization, Writing - review & editing, L.E. Murray: Conceptualization, Data curation, Formal analysis, Investigation, Methodology, Project administration, Resources, Software, Supervision, Validation, Visualization, Writing - original draft, Writing - review & editing, A.R. Popchock: Resources, Writing - review & editing, S. Biggins: Conceptualization, Funding acquisition, Project administration, Supervision, Writing - review & editing, C.L. Asbury: Conceptualization, Formal analysis, Funding acquisition, Investigation, Methodology, Resources, Software, Supervision, Writing - original draft, Writing - review & editing.

Disclosures: The authors declare no competing interests exist.

Submitted: 30 May 2024

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

# Supplemental material

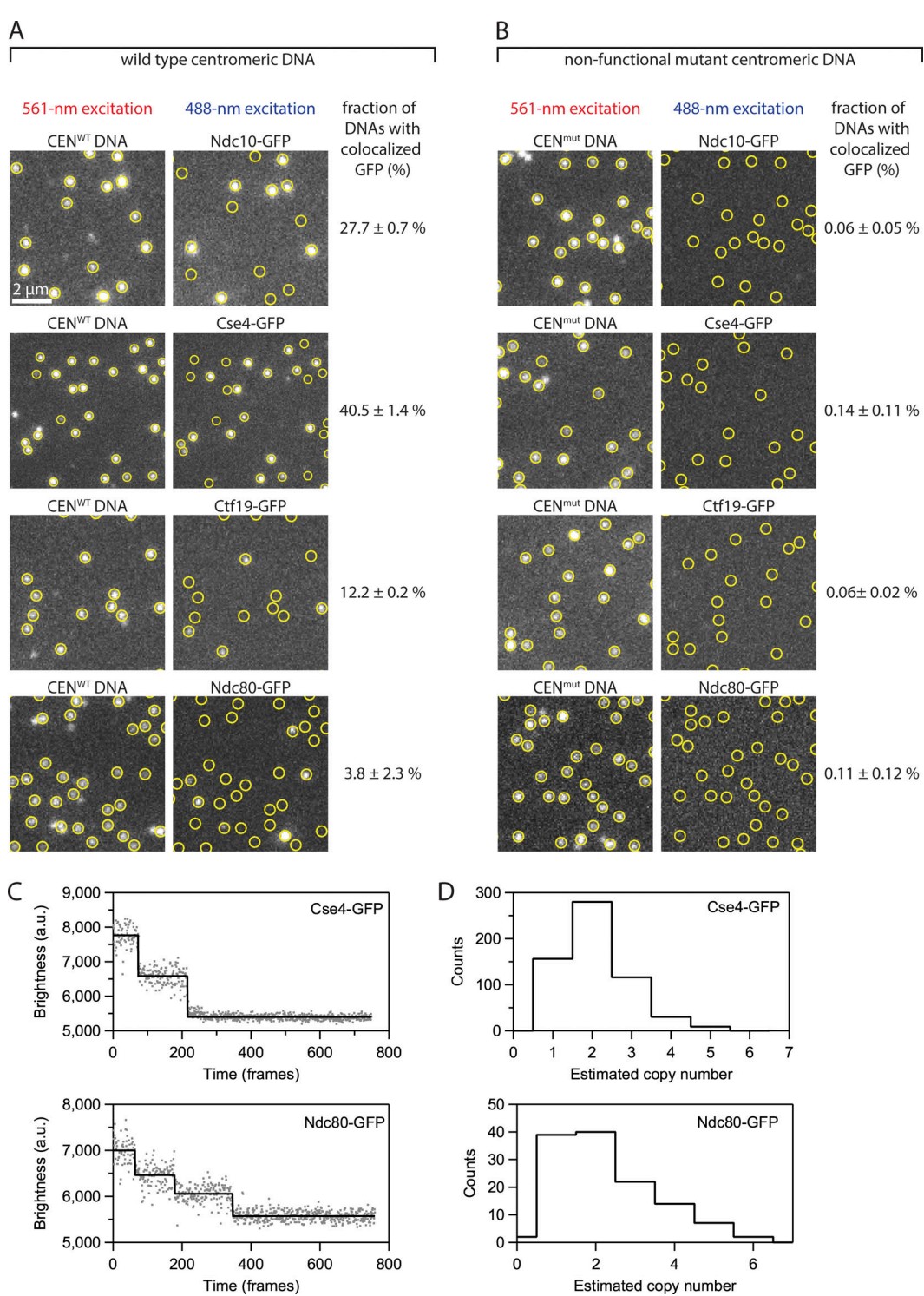

Figure S1.  **De novo assembly of individual kinetochores occurs specifically on centromeric DNAs. (A)** Images of individual Atto565-labeled centromeric DNAs (left) and corresponding images of GFP-tagged kinetochore subcomplexes (right) from the same fields of view. Yellow circles mark locations of individual wild type centromeric DNAs (CEN$^{WT}$), which contain the complete 117-bp centromere sequence from *S. cerevisiae* chromosome III. **(B)** Images from negative control experiments using mutant centromeric DNAs (CEN$^{mut}$) carrying a 3-bp substitution that prevents kinetochore assembly. The percentages in both A and B represent average fractions (±SEM) of centromeric DNAs that colocalized with a GFP signal from the indicated kinetochore protein, calculated from *N* > 3,400 DNAs for each kinetochore component from at least nine fields of view across three independent experiments. **(C)** Photobleach analysis of Cse4- or Ndc80-GFP assembled kinetochores. Representative records of fluorescence intensity versus time for individual Cse4-GFP (top) or Ndc80-GFP (bottom) assembled kinetochores. The raw intensity data is represented by gray spots and the estimated bleach steps are represented by the solid black line. Bleach steps were estimated using the Tdetector2 step detection algorithm (Chen et al., 2014). **(D)** Histograms showing the estimated copy number of Cse4-GFP (top) or Ndc80-GFP (bottom) present in individually assembled kinetochores. *N* = 591 individual Cse4-GFP kinetochore assemblies and *N* = 126 individual Ndc80-GFP kinetochore assemblies.

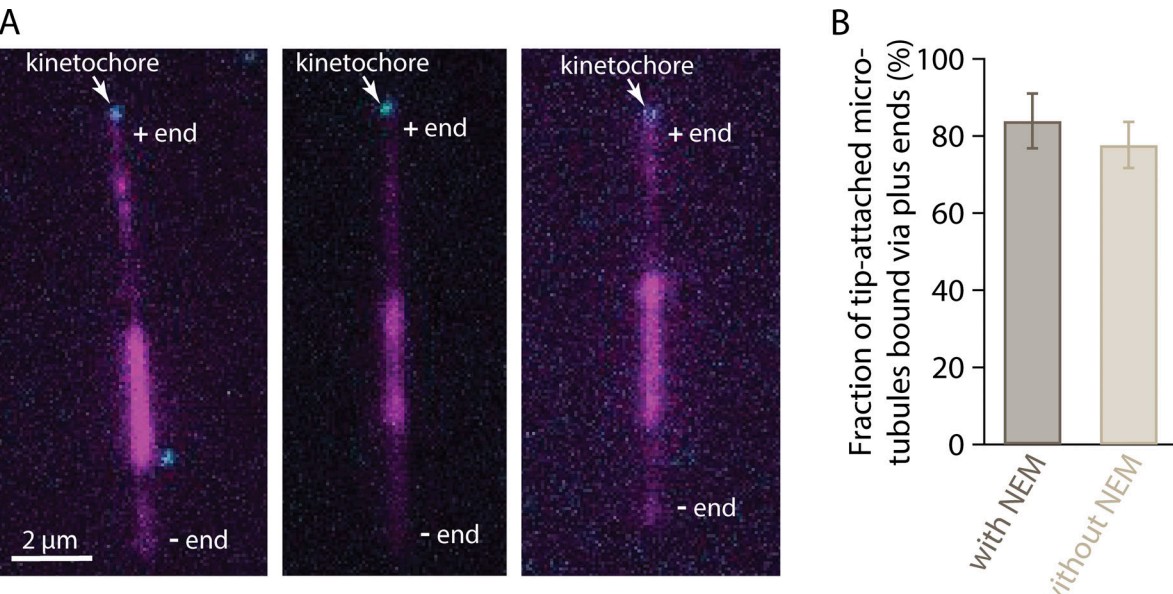

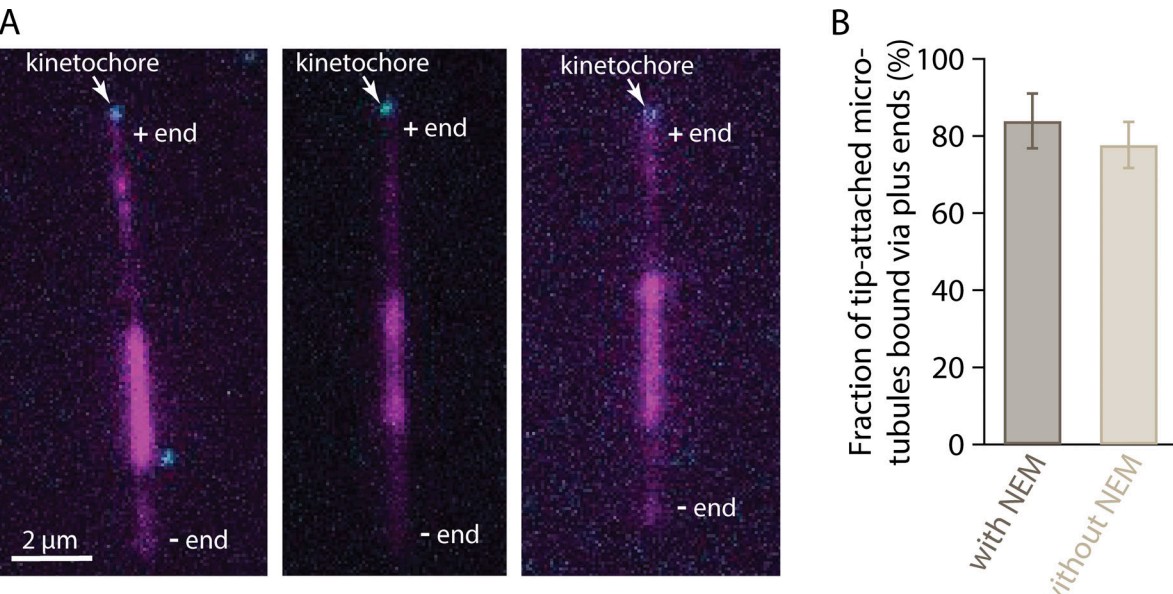

Figure S2.  **Plus-end preference is not an artifact of differential labeling. (A)** Polarity-marked microtubules (magenta), polymerized from bright seeds with dim extensions on both plus and minus ends, were nearly always captured via their plus ends by assembled Ndc80-GFP kinetochores (cyan). To polymerize dim extensions from both ends of bright seeds, NEM-treated tubulin was omitted from the polymerization mix (see Materials and methods). Polymerization at plus ends is faster than at minus ends, so plus ends were distinguishable by their longer lengths relative to minus ends. **(B)** Percentages of tip-captured, polarity-marked microtubules that were bound by their plus ends. A strong preference for plus ends occurred irrespective of whether the minus ends were more brightly labeled, via polymerization with a small amount of NEM-treated tubulin (with NEM, at left), or whether the minus and plus ends were both dimly labeled, via polymerization without NEM-treated tubulin (without NEM, at right). See Materials and methods for details about how polarity-marked microtubules were generated. Bars represent percentages of polarity-marked microtubules that were bound by their plus ends (mean ± SD, from $N = 4$ experiments with NEM and $N = 4$ without NEM, examining a total of 86 and 110 tip-captured microtubules, respectively).

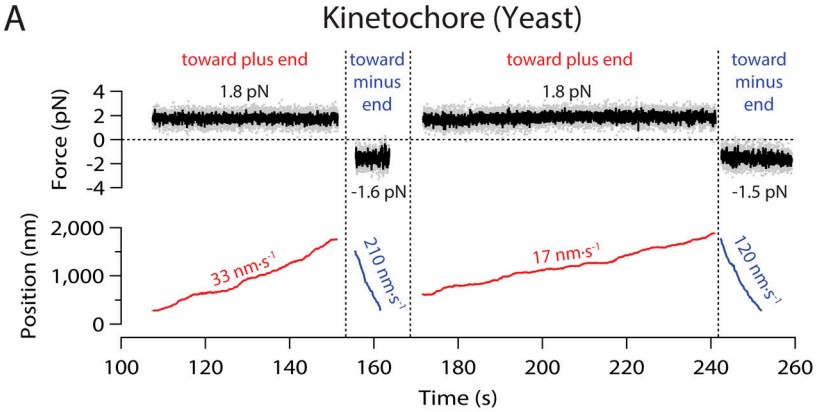

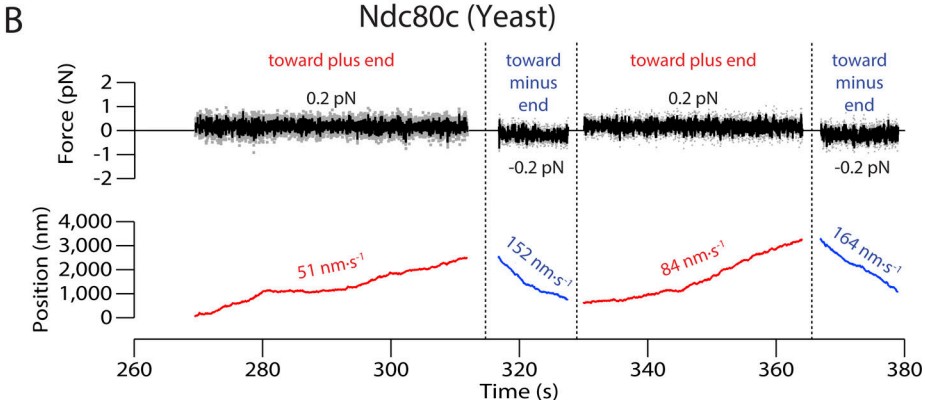

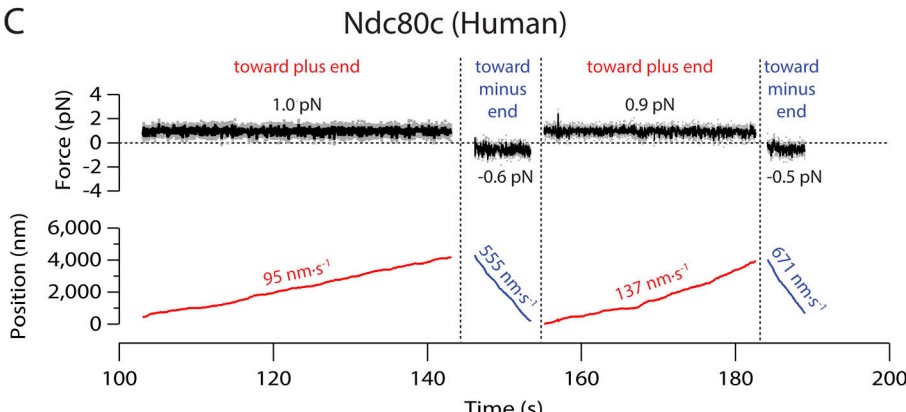

Figure S3.  **Example records showing measurement of bidirectional sliding friction. (A–C)** Force and position are plotted against time for beads decorated with (A) native yeast kinetochore particles, (B) recombinant yeast Ndc80c, or (C) recombinant human Ndc80c, attached to the sides of coverslip-anchored microtubules and pulled alternately toward the plus (red traces) and minus ends (blue traces). Mean forces and speeds for each sliding event are indicated.

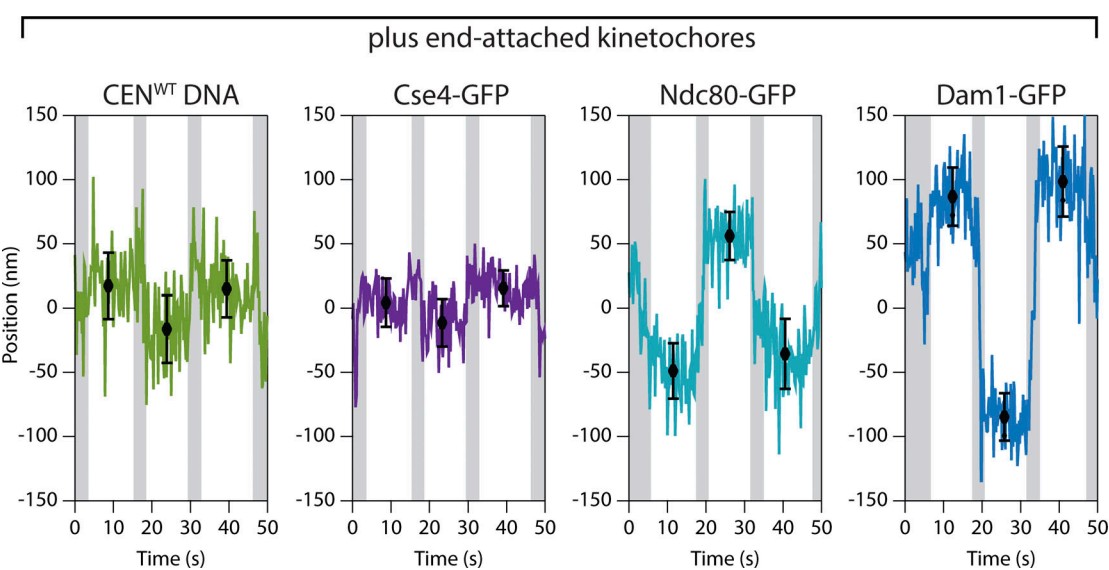

Figure S4. **Nanoscale displacement of fluorescent-tagged components within individual plus-end-attached kinetochore assemblies during periodic flow-induced reorientation.** Positions for the indicated GFP-labeled components were tracked with subpixel accuracy while the direction of fluid flow was oscillated, causing the kinetochore and its captured microtubule to flip back and forth, reorienting by 180° with each reversal of the flow. Displacements from the tether point were estimated by averaging during the intervals when the microtubule orientation was steady. Black symbols represent mean ± SD from $N$ = 60 tracked positions during each interval. Positions recorded during the reorientation of the microtubule were omitted from averaging and are indicated here by gray shading.

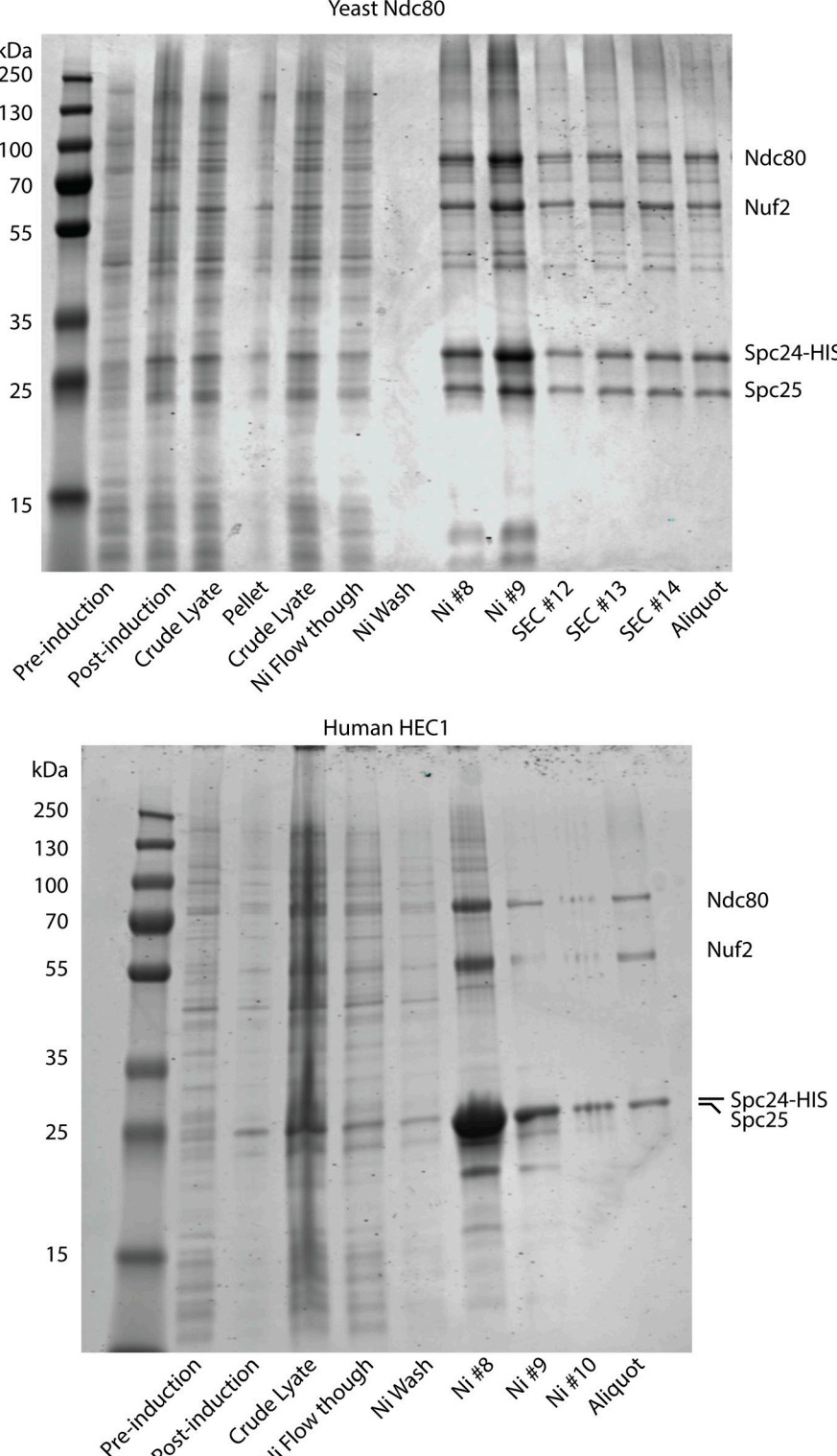

Figure S5.   **Coomassie-stained SDS-PAGE gels showing the recombinantly purified yeast and human Ndc80 complexes.** Source data are available for this figure: SourceData FS5. SEC indicates fractions collected after size exclusion chromatography.

Video 1.   **Time-lapse TIRF microscopy of assembled kinetochore with GFP-tagged Ndc80 (cyan) bound to the tip of an Alexa647-labeled taxol-stabilized microtubule (magenta) from** Fig. 1 D**.** Frames were collected every 500 ms. Video playback is 30 frames per second.

Video 2.   **Time-lapse TIRF microscopy of assembled kinetochore with GFP-tagged Ndc80 (cyan) bound to the side of an Alexa647-labeled taxol-stabilized microtubule (magenta) from** Fig. 1 D**.** Frames were collected every 500 ms. Video playback is 30 frames per second.

Video 3.   **Time-lapse TIRF microscopy of assembled kinetochore with a GFP-tagged Ndc80 (cyan) bound to the tip of an Alexa647-labeled taxol-stabilized microtubule (magenta) with flow-induced oscillation (left) from** Fig. 3 C**.** Zoom-in showing the oscillation of the Ndc80-GFP spot around the DNA tether point on the coverslip, the approximate position of which is indicated by the vertical yellow line (right). Frames were collected every 200 ms. Video playback is 60 frames per second.

Video 4.   **Time-lapse TIRF microscopy of assembled kinetochore with a GFP-tagged Ndc80 (cyan) bound to the side of an Alexa647-labeled taxol-stabilized microtubule (magenta) with flow-induced oscillation (left) from** Fig. 4 A**.** Zoom-in showing the oscillation of the Ndc80-GFP spot around the DNA tether point on the coverslip, the approximate position of which is indicated by the vertical yellow line (right). Frames were collected every 200 ms. Video playback is 60 frames per second.

**Provided online are Table S1 and Table S2. Table S1 shows all the individual measured values used to create the graphs in the paper. Table S2 lists primers, plasmids, and strains.**

