## [Peer Review File · The Journal of Cell Biology]

Kinetochores grip microtubules with directionally asymmetric strength

Joshua Larson, Natalie Heitkamp, Lucas Murray, Andrew Popchock, Sue Biggins, and Charles Asbury

Corresponding Author(s): Charles Asbury, University of Washington School of Medicine

Review Timeline:

Submission Date:	2024-05-30
Editorial Decision:	2024-07-12
Revision Received:	2024-08-27
Editorial Decision:	2024-09-18
Revision Received:	2024-09-25

Monitoring Editor: Arshad Desai

Scientific Editor: Dan Simon

Transaction Report:

DOI: <https://doi.org/10.1083/jcb.202405176>

July 12, 2024

Re: JCB manuscript #202405176

Dr. Charles L Asbury
University of Washington School of Medicine
Physiology & Biophysics
1959 NE Pacific St, HSB G-424
Box 357290
Seattle, Washington 98195-7290

Dear Dr. Asbury,

Thank you for submitting your manuscript entitled "Kinetochores grip microtubules with directionally asymmetric strength." The manuscript was assessed by three expert reviewers, whose comments are appended to this letter.

As you will see, Reviewers #1 and #2 are supportive of the work while Reviewer #3 is more critical. After considering their feedback, we would be interested in a revision that addresses the comments on improved characterization of the protein composition of the reconstituted kinetochores and better delineation of the relationship between the heterogenous reconstituted structures and purified complexes (including some analysis of the impact of purified complex concentration on beads on the observed behaviors). Please also address the other reviewer comments which are aimed at improving the clarity and broader impact of the work.

GENERAL GUIDELINES:

Text limits: Character count for an Article is < 40,000, not including spaces. Count includes title page, abstract, introduction, results, discussion, and acknowledgments. Count does not include materials and methods, figure legends, references, tables, or supplemental legends.

Figures: Articles may have up to 10 main text figures. Figures must be prepared according to the policies outlined in our Instructions to Authors, under Data Presentation, <https://jcb.rupress.org/site/misc/ifora.xhtml>. All figures in accepted manuscripts will be screened prior to publication.

Supplemental information: There are strict limits on the allowable amount of supplemental data. Articles may have up to 5 supplemental figures. Up to 10 supplemental videos or flash animations are allowed. A summary of all supplemental material should appear at the end of the Materials and methods section.

Please note that JCB now requires authors to submit Source Data used to generate figures containing gels and Western blots with all revised manuscripts. This Source Data consists of fully uncropped and unprocessed images for each gel/blot displayed in the main and supplemental figures. Since your paper includes cropped gel and/or blot images, please be sure to provide one Source Data file for each figure that contains gels and/or blots along with your revised manuscript files. File names for Source Data figures should be alphanumeric without any spaces or special characters (i.e., SourceDataF#, where F# refers to the associated main figure number or SourceDataFS# for those associated with Supplementary figures). The lanes of the gels/blots should be labeled as they are in the associated figure, the place where cropping was applied should be marked (with a box), and molecular weight/size standards should be labeled wherever possible. Source Data files will be made available to reviewers during evaluation of revised manuscripts and, if your paper is eventually published in JCB, the files will be directly linked to specific figures in the published article.

The typical timeframe for revisions is three to four months. While most universities and institutes have reopened labs and allowed researchers to begin working at nearly pre-pandemic levels, we at JCB realize that the lingering effects of the COVID-

19 pandemic may still be impacting some aspects of your work, including the acquisition of equipment and reagents. Therefore, if you anticipate any difficulties in meeting this aforementioned revision time limit, please contact us and we can work with you to find an appropriate time frame for resubmission. Please note that papers are generally considered through only one revision cycle, so any revised manuscript will likely be either accepted or rejected.

Thank you for this interesting contribution to Journal of Cell Biology. You can contact us at the journal office with any questions at cellbio@rockefeller.edu.

Sincerely,

Arshad Desai, PhD
Monitoring Editor
Journal of Cell Biology

Dan Simon, PhD
Scientific Editor
Journal of Cell Biology

Reviewer #1 (Comments to the Authors (Required)):

Asbury and colleagues use a combination of single molecule approaches (TIRF, as well as laser trapping experiments) to investigate the properties of yeast kinetochores, either assembled on CEN DNA in extracts, or using the established affinity purification with Dsn1. Extending recently published work (Popchock et al., EMBO J, 2023), the authors show that outer kinetochore components are recruited to CEN DNA from yeast extracts, albeit with relatively low efficiency. Interestingly, these kinetochore particles capture stabilized microtubules preferentially at their plus ends. Force trap experiments with polarity-marked MTs demonstrate higher rupture force at plus versus minus-end, and also differential drag forces depending on whether kinetochores are dragged laterally towards the plus- versus the minus end. This property is at least partially explained by the intrinsic properties of the Ndc80 complex. The authors furthermore use co-localization experiments at laterally versus end-on attached kinetochores in the TIRF assay, with a clever reversal of flow direction to apply forces in this setting. These experiments suggest different relative configurations of kinetochore components when dragged towards plus- versus minus ends.

Overall, I find this to be an interesting study with elegant single-molecule experiments that provide new insights into the properties of kinetochores. The authors provide an intuitive model for how stronger grip of the kinetochore towards the plus end may support bi-orientation in cells. The study as presented is already mature and well-developed and it should be a very good fit for the JCB. I have a number of points that should be addressed before publication, most of them concern additional explanations and data analysis.

Main points

- 1) Figure 1B, line 98: the authors should provide a short comment on the relatively low fraction of Ndc80-GFP positive kinetochores. I fully agree that the number is sufficient for analysis, but the difference between occupancy with Cse4 versus outer complexes such as Ndc80c is very noticeable. Is that a consequence of decreased affinity or does it indicate that microtubules may be required for more effective assembly of the outer kinetochore on CEN DNA?
- 2) Figure 1 D, line 116: I'm assuming that only Ndc80-GFP positive spots can capture MTs, is that correct? Or are there kinetochores without discernible Ndc80 signal that nevertheless bound microtubules, either laterally or end-on? It would be great if the authors could provide statistics on this important point.
- 3) Figure 1D and 2A, line 123: Is there any indication how tip-attachment was achieved? That is, could any conversion of lateral to end-on attachment be observed after introduction of the microtubules? Is there a biased diffusion towards the plus-end, possibly as a consequence of the preferred binding towards plus?
- 4) Figure 2A: The taxol stabilized microtubules used here lack the chemical features that distinguish plus and minus ends in dynamic microtubules, i.e. GTP cap versus GDP lattice etc. How do the authors envision plus-end recognition to occur?

5) line 202: The authors could comment on the stronger grip of human Ndc80c: Is that a consequence of using a mammalian tubulin substrate, better matching the human Ndc80c? Or inherently different binding affinities between yeast and human Ndc80c?

6) line 245: as a note of caution it should be conceded here that force in this experiment is imposed only by the flow acting on the static microtubules. Force generation by dynamic plus-ends with "real" tubulin polymerization/depolymerization may impose different configurations on kinetochore components.

7) Line 260 (Figure 4D): how many Ndc80 molecules make up the combined observed fluorescence here? In other words, what is the ensemble size whose position is determined relative to CEN?

8) line 294: an additional point could be that the longer lifetime of the correct lateral attachment may also make it more likely to catch a disassembling plus end.

Reviewer #2 (Comments to the Authors (Required)):

The manuscript by Larson et al., "Kinetochores grip microtubules with directionally asymmetric strength," investigates the properties with which kinetochores and kinetochore associated, microtubule binding complexes associate with both the ends and sides of microtubules. This is an interesting and relevant line of study, since there is little information regarding (1) how kinetochores bind differentially to plus vs. minus ends of microtubules, and (2) if kinetochore binding strength to the sides of microtubules is regulated.

The authors here combine their expertise in kinetochore assembly and reconstitution from yeast extracts with optical trapping to address these questions. Using these (and other) approaches, the authors demonstrate that yeast CEN DNA adhered to coverslip surfaces and incubated with yeast extract, which they previously found was sufficient to assemble Cse4 (CENP-A)/CCAN-containing kinetochore complexes, could assemble outer kinetochore components including NDC80C and Dam1C. These assemblages were competent to bind both the ends and sides of taxol-stabilized microtubules. Using polarity-marked microtubules, they found that the assembled kinetochores showed a strong preference for plus end vs. minus end microtubule binding, and the plus ends exhibited stronger attachment to the kinetochores compared to minus ends (consistent with an earlier study using isolated CHO cell mitotic chromosomes, as the authors point out). Although it is still unclear why this preference exists, the authors can conclude that it is not likely due to differences in tip structure, microtubule dynamics, or presence/absence of a GTP cap at plus vs. minus ends. The authors then move to the headliner experiments, analyzing the properties of lateral kinetochore-microtubule attachments. Using beads coated with kinetochores purified from budding yeast, the authors find that when beads bound to the sides of microtubules are pulled to the minus ends of microtubules vs. the plus ends, they grip the microtubule less tightly and their speed of sliding along the microtubule is faster. They go on to repeat this sliding assay using beads coated with purified yeast or human NDC80C, the protein complex recognized as the direct, force-transducing linkage between kinetochores and microtubules. They find that NDC80C-coated beads exhibit similar asymmetry. Finally, the authors investigate why this asymmetry exists - they hypothesize that it could involve the architecture/spatial organization of kinetochore components when "facing" the minus end vs. the plus end. To address this, they measure distances between various kinetochore components at plus end (end-on) attachments and lateral attachments. For end-on attachments, the intra-kinetochore distance measurements largely match previously published data, and in this study, they specifically find that NDC80 complexes are ~37 nm from the microtubule-kinetochore tether point. Interestingly, the NDC80 measurement data for laterally-attached kinetochores presents as bi-modal, with one population at ~39 nm from the tether point and one at ~18 nm, leading the authors to hypothesize that when laterally-attached kinetochores are facing the plus end, they are properly organized (similar to plus-end attachments), and when facing the minus ends, they are not. They test this by revisiting the bead pulling/sliding assay and confirm that when beads coated with kinetochores are pulled to the minus ends of microtubules, the NDC80 complexes are much closer to the tether point vs. when they are pulled to the plus ends. They conclude from these data that kinetochores moving along the side of a microtubule towards the minus end are likely unstable due to a lack of proper kinetochore organization. The authors then propose an interesting model for how this asymmetry may contribute to the correction of erroneous kinetochore-microtubule attachments, which will be important to test in the future.

This is a well-executed study that addresses important, outstanding questions, and should be of high interest to the field. While the work leads to many unanswered questions as well, in my opinion, the paper as it stands is a significant contribution.

Specific comments:

(1) It is not clear what the "tether point" is, molecularly, in Figures 3 and 4. Is this the biotin/avidin spot? And if so, how is it being visualized to measure the center of fluorescence?

(2) What is the distribution of the human NDC80 complexes on the beads? Do the authors have an estimate for the number of attached complexes? Are oligomers required for the observed results? It would be helpful to include this information.

(3) I found Fig 2B a bit confusing when first looking through the figures. The way it is presented, it looks like the microtubule is trapped at both ends. I think the figure is trying to convey that the bead is attached at either end, not both? (very minor point)

Reviewer #3 (Comments to the Authors (Required)):

In the manuscript entitled "Kinetochores grip microtubules with directionally asymmetric strength" Larson et al reconstitute protein complexes belonging to kinetochores using budding yeast lysates and purified centromeric DNA. In doing so, they obtain a heterogeneous population that contains a small number of particles recruiting inner kinetochore proteins, and a minority (less than 4%) that contain also some outer kinetochore proteins. No systematic analysis of the protein composition of these particles is reported, however the authors observe a sub-stoichiometric recruitment of the Ndc80 complex. It is unclear whether other microtubule-binding proteins of the outer kinetochore are present, and their amount. The authors then report some phenomenological studies with these protein particles. They observe that some of these particles capture microtubules by their plus ends, and that it is easier to drag a laterally bound particle along the microtubule lattice towards the minus end compared to the plus end. These observations are accompanied by analogous observations using beads coated with purified yeast and human Ndc80 complexes. Neither of these protein complexes are characterised in the manuscript in terms of their purity or stoichiometry of their attachment to the beads. No systematic comparison is made between isolated yeast Ndc80 and the heterogeneous kinetochore-containing particles in terms of their sliding, which prevents the understanding of whether the Ndc80 alone is sufficient to explain the observed effects.

This reviewer is not convinced that the study presents a significant mechanistic insight into an area of interest to a general audience. This manuscript could be better suited to a more specialised journal, provided that the authors report some mechanistic underpinnings of the effects they observe, such as molecular interactions that mediate the reported asymmetry of the microtubule attachment, and/or the importance of these interactions for cell physiology.

Dear Arshad and Dan,

Thank you for overseeing the review of our manuscript entitled, “Kinetochores grip microtubules with directionally asymmetric strength”. We are grateful for your time and effort, and for the constructive feedback from the reviewers. Below, please find our detailed, point-by-point responses. As you will see, we have made many changes to the manuscript to address their comments. We hope you will agree that the paper is substantially improved, and that you will consider it worthy of publication in JCB.

Sincerely,

Chip Asbury and Josh Larson

Reviewer #1 (Comments to the Authors (Required)):

Asbury and colleagues use a combination of single molecule approaches (TIRF, as well as laser trapping experiments) to investigate the properties of yeast kinetochores, either assembled on CEN DNA in extracts, or using the established affinity purification with Dsn1. Extending recently published work (Popchock et al., EMBO J, 2023), the authors show that outer kinetochore components are recruited to CEN DNA from yeast extracts, albeit with relatively low efficiency. Interestingly, these kinetochore particles capture stabilized microtubules preferentially at their plus ends. Force trap experiments with polarity-marked MTs demonstrate higher rupture force at plus versus minus-end, and also differential drag forces depending on whether kinetochores are dragged laterally towards the plus- versus the minus end. This property is at least partially explained by the intrinsic properties of the Ndc80 complex. The authors furthermore use co-localization experiments at laterally versus end-on attached kinetochores in the TIRF assay, with a clever reversal of flow direction to apply forces in this setting. These experiments suggest different relative configurations of kinetochore components when dragged towards plus- versus minus ends.

Overall, I find this to be an interesting study with elegant single-molecule experiments that provide new insights into the properties of kinetochores. The authors provide an intuitive model for how stronger grip of the kinetochore towards the plus end may support bi-orientation in cells. The study as presented is already mature and well-developed and it should be a very good fit for the JCB. I have a number of points that should be addressed before publication, most of them concern additional explanations and data analysis.

We sincerely thank the reviewer for their positive comments and their insightful and constructive points.

Main points

1) Figure 1B, line 98: the authors should provide a short comment on the relatively low fraction of Ndc80-GFP positive kinetochores. I fully agree that the number is sufficient for analysis, but the difference between occupancy with Cse4 versus outer complexes such as Ndc80c is very noticeable. Is that a consequence of decreased affinity or does it indicate that microtubules may be required for more effective assembly of the outer kinetochore on CEN DNA?

Kinetochore assembly is thought to be an ordered, hierarchical process where the binding of outer kinetochore components depends on the prior binding of inner components. We therefore suggest that the low occupancy of Ndc80 relative to Cse4 is probably limited by the kinetics of assembly in the extracts rather than by decreased affinity (i.e., probably limited by kinetics not energetics). In the revised manuscript, we have attempted to clarify this point in the first section of results (line 100).

We are working hard to improve the efficiency of assembly in the extracts, and we hope to report measurements of kinetics in the future. In particular, it seems likely that post-translational modifications supporting kinetochore assembly *in vivo* may be limited during the de novo assembly in extracts. Two phospho-mimetic substitutions on Dsn1 are known to help stabilize the extract-assembled kinetochores (as described in Lang 2018 *Elife*, ref [39]) and these were included in our work. We believe it is likely that additional post-translational modifications further help to stabilize kinetochore assembly *in vivo*. Identifying these modifications and either mimicking them or promoting their occurrence in the extract assemblies is a priority for future work.

Despite the relatively low percentages, a single field of view contained hundreds of DNAs and dozens of kinetochore particles with associated Ndc80, which was more than sufficient to support our primary conclusions about the preferential capture of plus ends, and about the molecular arrangement within plus end versus side attachments.

The idea that microtubule attachment might enhance kinetochore assembly is interesting. While we have not thoroughly tested this possibility, preliminary experiments with taxol-stabilized microtubules added into the extract did not appear to improve the efficiency. There are however many permutations of this experiment that we could try to further explore this hypothesis in the future.

2) Figure 1 D, line 116: I'm assuming that only Ndc80-GFP positive spots can capture MTs, is that correct? Or are there kinetochores without discernible Ndc80 signal that nevertheless bound microtubules, either laterally or end-on? It would be great if the authors could provide statistics on this important point.

This is a very interesting point that we have also wondered about, especially considering the prior evidence that chromosomal passenger proteins, Bir1 and Sli15, can form a linkage *in vitro* between centromeric DNA (via the centromere-binding CBF3 complex) and microtubules, independently of Ndc80 (Sandall 2006 *Cell*, ref [40]). When we assembled Ndc80-GFP kinetochores on wild type centromeric DNAs, the vast majority of captured microtubules had a colocalized Ndc80-GFP signal. Rarely, a captured microtubule appeared to lack Ndc80-GFP. These infrequent observations could potentially be due to bleaching, or GFPs that had not matured. Alternatively, they might represent capture via the chromosomal passenger proteins independently of Ndc80. Unfortunately, we did not observe them frequently enough to gather meaningful statistics. Nevertheless, in the revised manuscript we mention these rare events and their possible origins in the second section of Results (line 124).

In the future, we plan to repeat the microtubule capture assay using various mutant extracts, including extracts depleted of Ndc80, Dam1, Stu2 and Sli15, to examine the specific molecular dependencies of the capture behavior. For the present study, we chose to focus on kinetochores assembled in wild type (non-depleted) extracts with Ndc80-GFP. Our data show clearly that the assembled particles interact directly with the microtubules, since the Ndc80-GFP signals oscillate together with the microtubules during flow reversals.

3) Figure 1D and 2A, line 123: Is there any indication how tip-attachment was achieved? That is, could any conversion of lateral to end-on attachment be observed after introduction of the microtubules? Is there a biased diffusion towards the plus-end, possibly as a consequence of the preferred binding towards plus?

We thank the reviewer for this interesting question. Because we imaged the captured microtubules only after washing out excess unbound filaments, we did not directly observe the capture process. While this aspect of our experimental protocol was mentioned in the Methods, in retrospect we should have made it clearer in the main text. The revised version now includes a better description of the capture experiment in the second section of Results (line 113). We envision that capture occurs via a two-step process, where both plus and minus ends bind initially, and then the mechanically weaker minus end-attachments are preferentially lost due to viscous forces during the washout. This two-step model is consistent with the differential rupture strengths we measured at plus versus minus ends, and with prior observations using isolated CHO cell chromosomes, as described in ref [24]. In our revised manuscript, we now mention this two-step model in the Discussion (line 340).

On the timescale of our observations, we did not observe thermally driven diffusion of the microtubules relative to the assembled kinetochores. However, in a couple of instances, the viscous forces during flow caused an initially side-attached microtubule to slide and apparently convert to an end-attached configuration or detach completely. Unfortunately, we did not observe this conversion often enough to study it carefully, but we hope it might become possible to study these conversions with further improvements in our technique.

4) Figure 2A: The taxol stabilized microtubules used here lack the chemical features that distinguish plus and minus ends in dynamic microtubules, i.e. GTP cap versus GDP lattice etc. How do the authors envision plus-end recognition to occur?

We hypothesize that all three of the intrinsic kinetochore behaviors we uncovered here – their preference for capturing (i) and holding (ii) microtubule plus ends under tension, and their directionally asymmetric grip when side-attached (iii) – all arise from the structural polarity of the microtubule and how it influences kinetochore architecture. At a plus end, the stalks of multiple Ndc80c fibrils can project past the tip of the microtubule to converge onto the centromeric nucleosome, potentially allowing Dam1c oligomers to organize a cage-like arrangement surrounding the tip. The formation of such a cage-like arrangement would substantially increase the interaction energy and thereby increase the strength of a plus end attachment beyond what is achievable at other locations on the microtubule where the cage cannot form. We have attempted to explain this hypothesis in the Discussion (lines 326 through 332 and 347 through 350) and to diagram it in Figures 5B and 5C.

5) line 202: The authors could comment on the stronger grip of human Ndc80c: Is that a consequence of using a mammalian tubulin substrate, better matching the human Ndc80c? Or inherently different binding affinities between yeast and human Ndc80c?

Preliminary measurements suggest that human Ndc80c forms attachments that are similar in strength, irrespective of whether the microtubules are assembled from mammalian (bovine) or yeast tubulin. We therefore favor the hypothesis of inherent strength differences between the human and yeast Ndc80c. In the future, we hope to dissect the molecular underpinnings, but in our view this goal falls outside the scope of the present study.

6) line 245: as a note of caution it should be conceded here that force in this experiment is imposed only by the flow acting on the static microtubules. Force generation by dynamic plus-ends with "real" tubulin polymerization/depolymerization may impose different configurations on kinetochore components.

We agree with the reviewer's note of caution here and we have edited the Results (line 251) to acknowledge this possibility.

7) Line 260 (Figure 4D): how many Ndc80 molecules make up the combined observed fluorescence here? In other words, what is the ensemble size whose position is determined relative to CEN?

Our best estimate of the ensemble size is given in Supplemental Figure S1D.

8) line 294: an additional point could be that the longer lifetime of the correct lateral attachment may also make it more likely to catch a disassembling plus end.

This is an excellent point and we have added it to the Discussion (line 307). We are grateful to the reviewer for this insightful comment.

Reviewer #2 (Comments to the Authors (Required)):

The manuscript by Larson et al., "Kinetochores grip microtubules with directionally asymmetric strength," investigates the properties with which kinetochores and kinetochore associated, microtubule binding complexes associate with both the ends and sides of microtubules. This is an interesting and relevant line of study, since there is little information regarding (1) how kinetochores bind differentially to plus vs. minus ends of microtubules, and (2) if kinetochore binding strength to the sides of microtubules is regulated.

The authors here combine their expertise in kinetochore assembly and reconstitution from yeast extracts with optical trapping to address these questions. Using these (and other) approaches, the authors demonstrate that yeast CEN DNA adhered to coverslip surfaces and incubated with yeast extract, which they previously found was sufficient to assemble Cse4 (CENP-A)/CCAN-containing kinetochore complexes, could assemble outer kinetochore components including NDC80C and Dam1C. These assemblages were competent to bind both the ends and sides of taxol-stabilized microtubules. Using polarity-marked microtubules, they found that the assembled kinetochores showed a strong preference for plus end vs. minus end microtubule binding, and the plus ends exhibited stronger attachment to the kinetochores compared to minus ends (consistent with an earlier study using isolated CHO cell mitotic chromosomes, as the authors point out). Although it is still unclear why this preference exists, the authors can conclude that it is not likely due to differences in tip structure, microtubule dynamics, or presence/absence of a GTP cap at plus vs. minus ends. The authors then move to the headliner experiments, analyzing the properties of lateral kinetochore-microtubule attachments. Using beads coated with kinetochores purified from budding yeast, the authors find that when beads bound to the sides of microtubules are pulled to the minus ends of microtubules vs. the plus ends, they grip the microtubule less tightly and their speed of sliding along the microtubule is faster. They go on to repeat this sliding assay using beads coated with purified yeast or human NDC80C, the protein complex recognized as the direct, force-transducing linkage between kinetochores and microtubules. They find that NDC80C-coated beads exhibit similar asymmetry. Finally, the authors investigate why this

asymmetry exists - they hypothesize that it could involve the architecture/spatial organization of kinetochore components when "facing" the minus end vs. the plus end. To address this, they measure distances between various kinetochore components at plus end (end-on) attachments and lateral attachments. For end-on attachments, the intra-kinetochore distance measurements largely match previously published data, and in this study, they specifically find that NDC80 complexes are ~37 nm from the microtubule-kinetochore tether point. Interestingly, the NDC80 measurement data for laterally-attached kinetochores presents as bi-modal, with one population at ~39 nm from the tether point and one at ~18 nm, leading the authors to hypothesize that when laterally-attached kinetochores are facing the plus end, they are properly organized (similar to plus-end attachments), and when facing the minus ends, they are not. They test this by revisiting the bead pulling/sliding assay and confirm that when beads coated with kinetochores are pulled to the minus ends of microtubules, the NDC80 complexes are much closer to the tether point vs. when they are pulled to the plus ends. They conclude from these data that kinetochores moving along the side of a microtubule towards the minus end are likely unstable due to a lack of proper kinetochore organization. The authors then propose an interesting model for how this asymmetry may contribute to the correction of erroneous kinetochore-microtubule attachments, which will be important to test in the future.

This is a well-executed study that addresses important, outstanding questions, and should be of high interest to the field. While the work leads to many unanswered questions as well, in my opinion, the paper as it stands is a significant contribution.

We sincerely thank the reviewer for their positive comments and their constructive points.

Specific comments:

(1) It is not clear what the "tether point" is, molecularly, in Figures 3 and 4. Is this the biotin/avidin spot? And if so, how is it being visualized to measure the center of fluorescence?

We apologize for this confusion. Yes, the 'tether point' is the biotin-avidin linkage that anchors the DNA to the PEG/Biotin-PEG passivated coverslip. We did not directly observe the tether point. Rather, we inferred its position as the midpoint between the tracked positions of the fluorescent-tagged kinetochore components before and after each flow reversal. In the revised manuscript, we modified the schematics of Figures 3 and 4 to explicitly show the tether point, and we added an explanation of how the tether point was inferred to the legend of Figure 3D.

(2) What is the distribution of the human NDC80 complexes on the beads? Do the authors have an estimate for the number of attached complexes? Are oligomers required for the observed results? It would be helpful to include this information.

We thank the reviewer for this important and interesting question. For our experiments that used recombinant yeast or human Ndc80c, the beads were densely coated, such that each bead was decorated with ~3,000 complexes. Based on simple geometric considerations (detailed in Hamilton 2020 *Elife*, ref [50]), we estimate that a maximum of ~90 Ndc80 complexes would be capable of simultaneously binding the microtubule surface under the conditions of our sliding friction measurements. Thus oligomerization (a.k.a. 'clustering') of Ndc80c was indeed possible. In retrospect, we failed to make this important point clear in our original submission. The revised Methods section now includes a much more detailed description of the preparation of Ndc80c-coated beads, with

estimated complex-to-bead ratios (line 640). In the main text Results, we now explicitly mention that the experiments with recombinant Ndc80c were not conducted under single-molecule conditions (line 201). The interesting possibility that oligomerization of Ndc80 complexes might contribute to mechanical asymmetry, particularly in humans which lack Dam1c, is now mentioned in the Discussion, along with a citation of the recent work by Polley et al (2023 *EMBO J*, ref [60]) showing that loop-dependent clustering strengthens the human Ndc80c-microtubule interface (line 332).

(3) I found Fig 2B a bit confusing when first looking through the figures. The way it is presented, it looks like the microtubule is trapped at both ends. I think the figure is trying to convey that the bead is attached at either end, not both? (very minor point)

We thank the reviewer for noting this. We agree that the original Figure 2B was a bit confusing and we have now revised it to better depict how the experiment was conducted.

Reviewer #3 (Comments to the Authors (Required)):

In the manuscript entitled "Kinetochores grip microtubules with directionally asymmetric strength" Larson et al reconstitute protein complexes belonging to kinetochores using budding yeast lysates and purified centromeric DNA. In doing so, they obtain a heterogeneous population that contains a small number of particles recruiting inner kinetochore proteins, and a minority (less than 4%) that contain also some outer kinetochore proteins. No systematic analysis of the protein composition of these particles is reported, however the authors observe a sub-stoichiometric recruitment of the Ndc80 complex. It is unclear whether other microtubule-binding proteins of the outer kinetochore are present, and their amount.

The kinetochore particles were assembled in yeast cell extracts following essentially the same methods as reported previously (Lang 2018 *Elife*, ref [39]) except that the centromeric DNAs were attached sparsely to coverslips rather than densely to magnetic beads. The previous work included what we consider to be a systematic analysis of the levels of many kinetochore proteins that co-assembled onto the DNA, examined by Western blotting and mass spectrometry. In addition to the Ndc80 complex, the Dam1 complex, Stu2, and also the chromosomal passenger complex, were clearly detectable.

As mentioned above in our responses to reviewer #1, kinetochore assembly is thought to be an ordered, hierarchical process where the binding of outer kinetochore components depends on the prior binding of inner components. We therefore suggest that the low occupancy of Ndc80 relative to Cse4 is probably limited by the kinetics of assembly in the extracts rather than by decreased affinity (i.e., probably limited by kinetics not energetics). In the revised manuscript, we have attempted to clarify this point in the first section of results (line 100).

We are working hard to improve the efficiency of assembly in the extracts, and we hope to report measurements of kinetics in the future. In particular, it seems likely that post-translational modifications supporting kinetochore assembly *in vivo* may be limited during the *de novo* assembly in extracts. Two phospho-mimetic substitutions on Dsn1 are known to help stabilize the extract-assembled kinetochores (as described in Lang 2018 *Elife*, ref [39]) and these were included in our work. We believe it is likely that additional post-translational modifications further help to stabilize kinetochore assembly *in vivo*. Identifying these modifications and either mimicking them or promoting their occurrence in the extract assemblies is a priority for future work.

Despite the relatively low percentages, a single field of view contained hundreds of DNAs and dozens of kinetochore particles with associated Ndc80, which was sufficient to support our primary conclusions about the preferential capture of plus ends, and about the molecular arrangement within plus end versus side attachments.

The authors then report some phenomenological studies with these protein particles. They observe that some of these particles capture microtubules by their plus ends, and that it is easier to drag a laterally bound particle along the microtubule lattice towards the minus end compared to the plus end. These observations are accompanied by analogous observations using beads coated with purified yeast and human Ndc80 complexes. Neither of these protein complexes are characterised in the manuscript in terms of their purity or stoichiometry of their attachment to the beads.

We sincerely apologize that our original manuscript did not include evidence supporting the purity of our Ndc80c preparations, and lacked sufficient detail about the density of the complexes on the microbeads. The yeast and human Ndc80 (Hec1) complexes were purified essentially as described previously (Hamilton 2020 *Elife*, ref [50]; Helgeson 2018 *PNAS*, ref [51]). The revised Methods now includes a much more detailed description of how the Ndc80c-coated beads were prepared, with estimated complex-to-bead ratios (line 640). In the main text Results, we now explicitly mention that the experiments with recombinant Ndc80c were not conducted under single-molecule conditions (line 201). Images of Coomassie-stained SDS-PAGE analyses have been added (Figure S5) to illustrate the purity of the preparations.

No systematic comparison is made between isolated yeast Ndc80 and the heterogeneous kinetochore-containing particles in terms of their sliding, which prevents the understanding of whether the Ndc80 alone is sufficient to explain the observed effects.

Like the reviewer, we are deeply interested in understanding the mechanistic origins of the strongly asymmetric sliding behavior of native yeast kinetochore particles. The fact that Ndc80c alone exhibits qualitatively similar behavior suggests it is at least partly responsible for the asymmetry of the native particles. But as we note in our Discussion (line 318), the yeast Ndc80c alone is relatively weak in comparison to the native particles, and the human Ndc80c alone showed less dramatic mechanical asymmetry than the native yeast kinetochore particles. It therefore seems likely to us that additional kinetochore subcomplexes contribute. Much more work will be needed to systematically determine which specific subcomplexes contribute, and the degree to which they contribute. For example, in the future we plan to repeat the sliding friction measurements using kinetochore particles isolated from various mutant strains. But in our view, such experiments go beyond the scope of the present study. The highly direction-sensitive grip of the kinetochore is well supported by our data, and the qualitatively similar behavior of Ndc80c alone already suggests it makes a substantial contribution.

This reviewer is not convinced that the study presents a significant mechanistic insight into an area of interest to a general audience. This manuscript could be better suited to a more specialised journal, provided that the authors report some mechanistic underpinnings of the effects they observe, such as molecular interactions that mediate the reported asymmetry of the microtubule attachment, and/or the importance of these interactions for cell physiology.

With respect, we disagree with the reviewer's assessment. The strongly asymmetric grip that we uncovered may allow kinetochores to distinguish correct from incorrect side-attachments during early

mitosis *in vivo*. This is a concept that to our knowledge has not previously been considered and seems likely to be of wide interest to cell biologists. In mitosis, it can help to explain how sister kinetochores selectively attach microtubules emanating from opposite poles with astounding accuracy. More generally, asymmetric gripping may explain how cytoskeletal junctions self-assemble with appropriately oriented filaments.

September 18, 2024

RE: JCB Manuscript #202405176R

Dr. Charles L Asbury
University of Washington School of Medicine
Physiology & Biophysics
1959 NE Pacific St, HSB G-424
Box 357290
Seattle, Washington 98195-7290

Dear Dr. Asbury:

Thank you for submitting your revised manuscript entitled "Kinetochores grip microtubules with directionally asymmetric strength." We would be happy to publish your paper in JCB pending final revisions necessary to meet our formatting guidelines (see details below).

A. MANUSCRIPT ORGANIZATION AND FORMATTING:

1) Text limits: Character count for Articles is < 40,000, not including spaces. Count includes title page, abstract, introduction, results, discussion, and acknowledgments. Count does not include materials and methods, figure legends, references, tables, or supplemental legends.

2) Figure formatting: Articles may have up to 10 main text figures. Scale bars must be present on all microscopy images, including inset magnifications. Molecular weight or nucleic acid size markers must be included on all gel electrophoresis. Please add a scale bar to Figure S2A.

Also, please avoid pairing red and green for images and graphs to ensure legibility for color-blind readers. If red and green are paired for images, please ensure that the particular red and green hues used in micrographs are distinctive with any of the colorblind types. If not, please modify colors accordingly or provide separate images of the individual channels.

3) Statistical analysis: Error bars on graphic representations of numerical data must be clearly described in the figure legend. The number of independent data points (n) represented in a graph must be indicated in the legend. Please, indicate whether 'n' refers to technical or biological replicates (i.e. number of analyzed cells, samples or animals, number of independent experiments). If independent experiments with multiple biological replicates have been performed, we recommend using distribution-reproducibility SuperPlots (please see Lord et al., JCB 2020) to better display the distribution of the entire dataset, and report statistics (such as means, error bars, and P values) that address the reproducibility of the findings.

Statistical methods should be explained in full in the materials and methods. For figures presenting pooled data the statistical measure should be defined in the figure legends. Please also be sure to indicate the statistical tests used in each of your experiments (both in the figure legend itself and in a separate methods section) as well as the parameters of the test (for example, if you ran a t-test, please indicate if it was one- or two-sided, etc.). Also, if you used parametric tests, please indicate if the data distribution was tested for normality (and if so, how). If not, you must state something to the effect that "Data distribution was assumed to be normal but this was not formally tested."

4) Materials and methods: Should be comprehensive and not simply reference a previous publication for details on how an experiment was performed. Please provide full descriptions (at least in brief) in the text for readers who may not have access to referenced manuscripts. The text should not refer to methods "...as previously described."

5) For all cell lines, vectors, constructs/cDNAs, etc. - all genetic material: please include database / vendor ID (e.g., Addgene, ATCC, etc.) or if unavailable, please briefly describe their basic genetic features, even if described in other published work or gifted to you by other investigators (and provide references where appropriate). Please be sure to provide the sequences for all of your oligos: primers, si/shRNA, RNAi, gRNAs, etc. in the materials and methods. You must also indicate in the methods the source, species, and catalog numbers/vendor identifiers (where appropriate) for all of your antibodies, including secondary. If antibodies are not commercial, please add a reference citation if possible.

6) Microscope image acquisition: The following information must be provided about the acquisition and processing of images:
a. Make and model of microscope
b. Type, magnification, and numerical aperture of the objective lenses

- c. Temperature
- d. Imaging medium
- e. Fluorochromes
- f. Camera make and model
- g. Acquisition software
- h. Any software used for image processing subsequent to data acquisition. Please include details and types of operations involved (e.g., type of deconvolution, 3D reconstitutions, surface or volume rendering, gamma adjustments, etc.).

7) References: There is no limit to the number of references cited in a manuscript. References should be cited parenthetically in the text by author and year of publication. Abbreviate the names of journals according to PubMed.

8) Supplemental materials: Articles may have up to 5 supplemental figures and 10 videos.

Please also note that tables, like figures, should be provided as individual, editable files. A summary of all supplemental material should appear at the end of the Materials and methods section. Please include one brief sentence per item.

9) Video legends: Should describe what is being shown, the cell type or tissue being viewed (including relevant cell treatments, concentration and duration, or transfection), the imaging method (e.g., time-lapse epifluorescence microscopy), what each color represents, how often frames were collected, the frames/second display rate, and the number of any figure that has related video stills or images.

10) eTOC summary: A ~40-50 word summary that describes the context and significance of the findings for a general readership should be included on the title page. The statement should be written in the present tense and refer to the work in the third person. It should begin with "First author name(s) et al..." to match our preferred style.

11) Conflict of interest statement: JCB requires inclusion of a statement in the acknowledgements regarding competing financial interests. If no competing financial interests exist, please include the following statement: "The authors declare no competing financial interests." If competing interests are declared, please follow your statement of these competing interests with the following statement: "The authors declare no further competing financial interests."

12) A separate author contribution section is required following the Acknowledgments in all research manuscripts. All authors should be mentioned and designated by their first and middle initials and full surnames. We encourage use of the CRediT nomenclature (<https://casrai.org/credit/>).

13) ORCID IDs: ORCID IDs are unique identifiers allowing researchers to create a record of their various scholarly contributions in a single place. Please note that ORCID IDs are required for all authors. At resubmission of your final files, please be sure to provide your ORCID ID and those of all co-authors.

14) JCB requires authors to submit Source Data used to generate figures containing gels and Western blots with all revised manuscripts. This Source Data consists of fully uncropped and unprocessed images for each gel/blot displayed in the main and supplemental figures. Since your paper includes cropped gel and/or blot images, please be sure to provide one Source Data file for each figure that contains gels and/or blots along with your revised manuscript files. File names for Source Data figures should be alphanumeric without any spaces or special characters (i.e., SourceDataF#, where F# refers to the associated main figure number or SourceDataFS# for those associated with Supplementary figures). The lanes of the gels/blots should be labeled as they are in the associated figure, the place where cropping was applied should be marked (with a box), and molecular weight/size standards should be labeled wherever possible. Source Data files will be directly linked to specific figures in the published article.

15) Journal of Cell Biology now requires a data availability statement for all research article submissions. These statements will be published in the article directly above the Acknowledgments. The statement should address all data underlying the research presented in the manuscript. Please visit the JCB instructions for authors for guidelines and examples of statements at (<https://rupress.org/jcb/pages/editorial-policies#data-availability-statement>).

B. FINAL FILES:

-- High-resolution figure and MP4 video files: See our detailed guidelines for preparing your production-ready images,

<https://jcb.rupress.org/fig-vid-guidelines>.

Thank you for your attention to these final processing requirements. Please revise and format the manuscript and upload materials within 7 days. If you need an extension for whatever reason, please let us know and we can work with you to determine a suitable revision period.

Thank you for this interesting contribution, we look forward to publishing your paper in Journal of Cell Biology.

Sincerely,

Arshad Desai, PhD
Monitoring Editor
Journal of Cell Biology

Dan Simon, PhD
Scientific Editor
Journal of Cell Biology

Reviewer #1 (Comments to the Authors (Required)):

In this revised version the authors have answered my questions and comments to my full satisfaction. The edits to the manuscripts have clarified various points and further improved what was already a very careful and interesting study. I am happy to support publication of the manuscript in the JCB and congratulate the authors on their interesting work.